# 1 Classifying Thermodynamic Cloud Phase Using Machine Learning

# 2 Models

- 3 Lexie Goldberger<sup>1\*</sup>, Maxwell Levin<sup>1\*</sup>, Carlandra Harris<sup>1,2</sup>, Andrew Geiss<sup>1</sup>, Matthew D. Shupe<sup>3,4</sup>, Damao
- 4 Zhang<sup>1</sup>
- 5 Pacific Northwest National Laboratory, Richland, 99352, USA
- 6 <sup>2</sup> Alabama State University, Montgomery, 36104, USA
- <sup>3</sup> Cooperative Institute for Research in Environmental Sciences, University of Colorado Boulder, Boulder, CO, USA
- <sup>4</sup> Physical Sciences Laboratory, National Oceanic and Atmospheric Administration, Boulder, CO, USA
- <sup>\*</sup> These authors contributed equally to this work.
- 10 Correspondence to: Damao Zhang (Damao.Zhang@pnnl.gov)

Abstract. Vertically resolved thermodynamic cloud phase classifications are essential for studies of atmospheric cloud and 11 12 precipitation processes. The Department of Energy (DOE) Atmospheric Radiation Measurement (ARM) 13 THERMOCLDPHASE Value-Added Product (VAP) uses a multi-sensor approach to classify thermodynamic cloud phase by combining lidar backscatter and depolarization, radar reflectivity, Doppler velocity, spectral width, microwave radiometer-14 15 derived liquid water path, and radiosonde temperature measurements. The measured pixels are classified as ice, snow, mixedphase, liquid (cloud water), drizzle, rain, and liq driz (liquid+drizzle). We use this product as the ground truth to train three 16 17 machine learning (ML) models to predict the thermodynamic cloud phase from multi-sensor remote sensing measurements 18 taken at the ARM North Slope of Alaska (NSA) observatory: a random forest (RF), a multilayer perceptron (MLP), and a 19 convolutional neural network (CNN) with a U-Net architecture. Evaluations against the outputs of the THERMOCLDPHASE 20 VAP with one year of data show that the CNN outperforms the other two models, achieving the highest test accuracy, F1-21 score, and mean Intersection over Union (IOU). Analysis of ML confidence scores shows ice, rain, and snow have higher 22 confidence scores, followed by liquid, while mixed, drizzle, and liq driz have lower scores. Feature importance analysis 23 reveals that the mean Doppler velocity and vertically resolved temperature are the most influential datastreams for ML 24 thermodynamic cloud phase predictions. Lidar measurements exhibit lower feature importance due to rapid signal attenuation 25 caused by the frequent presence of persistent low-level clouds at the NSA site. The ML models' generalization capacity is 26 further evaluated by applying them at another Arctic ARM site in Norway using data taken during the ARM Cold-Air 27 Outbreaks in the Marine Boundary Layer Experiment (COMBLE) field campaign. The models demonstrated similar 28 performance to that observed at the NSA site. Finally, we evaluate the ML models' response to simulated instrument outages 29 and signal degradation and show that a CNN U-NET model trained with input channel dropouts performs better when input 30 fields are missing.

#### 1 Introduction

- 32 Arctic clouds are one of the least understood elements of the Arctic climate system, but they play a significant role in regulating 33 radiative energy fluxes at the surface, through the atmosphere, and at the top of the atmosphere (Cesana & Chepfer, 2012; 34 Curry et al., 1996; Kay & L'Ecuyer, 2013; Kay et al., 2008; Shupe & Intrieri, 2004). One major factor in this uncertainty is the 35 thermodynamic phase of clouds, which is crucial for understanding many cloud processes, including ice particle production via the Wegener-Bergeron-Findeisen process (Storelymo and Tan 2015), precipitation formation (Mülmenstädt et al., 2015). 36 37 the evolution of the cloud life cycle (Pithan et al., 2014), and also the response of clouds to global warming (Tan et al., 2025). 38 Ice particles and liquid droplets differ significantly in number, size, shape, fall velocity, and refractive index, leading to vastly 39 different radiative properties for clouds with different thermodynamic structures (Shupe & Intrieri, 2004). Accurate 40 thermodynamic cloud phase representations in climate models enhance the reliability of climate projections (Cesana et al., 41 2024). In addition, thermodynamic cloud phase classification is often a prerequisite for retrieving cloud properties from 42 remote-sensing data, as most retrieval algorithms are designed for specific thermodynamic cloud phases and types (Shupe et 43 al., 2015; Platnick et al., 2017). 44 Thermodynamic cloud phase can be determined using either aircraft in situ measurements (McFarquhar et al., 2011; Verlinde 45 et al., 2007; Wendisch et al., 2019) or remote sensing observations (Avery et al., 2020; Barker et al., 2008; Hogan et al., 2003; 46 Shupe, 2007; Turner et al., 2003). Aircraft in situ measurements use captured particle images from onboard probes to identify 47 thermodynamic cloud phase based on the shape and size of cloud particles. While in situ measurements offer thermodynamic 48 cloud phase identification, it is challenging to gather large aircraft datasets under diverse environmental conditions, and these 49 measurements cannot provide routine or continuous daily data. Remote sensing observations, however, offer long-term 50 continuous thermodynamic cloud phase identification. Space-borne remote sensing, in particular, enables global-scale 51 thermodynamic cloud phase classification, which can effectively constrain global climate models (Cesana & Chepfer, 2013; 52 Tan et al., 2016). High-resolution ground-based remote sensing observations allow for detailed thermodynamic cloud 53 classification, supporting studies of cloud processes, and the validation of high-resolution cloud-resolving model simulations 54 (Fan et al., 2011; Kalesse et al., 2016). 55 The Department of Energy (DOE) Atmospheric Radiation Measurement (ARM) user facility deploys advanced remote sensing 56 instruments in climate-critical locations to monitor atmospheric states and processes. To address the need for accurate 57 thermodynamic cloud phase identification, ARM developed the Thermodynamic Cloud Phase (THERMOCLDPHASE) 58 Value-Added Product (VAP) (Zhang & Levin, 2024). Using the multi-sensor approach developed by Shupe (2007), the 59 THERMOCLDPHASE VAP combines data from active remote-sensing lidars, radars, microwave radiometers, and 60 radiosondes to determine vertically resolved thermodynamic cloud phase at ARM sites. THERMOCLDPHASE data are 61 available through ARM Data Discovery for ARM's North Slope of Alaska (NSA) atmospheric observatory at Utqiagvik,
- Alaska from 2018 to 2022, as well as six other ARM high-latitude observatories ("ARM Data Discovery," 2025). It is noted 62 63

- clouds (Shupe 2007). Since the algorithm does not include the classification of hail and graupel, it has difficulties
- distinguishing these hydrometeor types in deep convective cloud regimes over tropical and mid-latitude regions.
- Threshold-based algorithms for determining the thermodynamic cloud phase, such as those used in the THERMOCLDPHASE
- VAP, have two major limitations. First, standard algorithms are static and do not improve with additional data or generalize
- to new regions. For ARM to apply the Shupe (2007) algorithm to sites other than the Arctic, where it was originally developed,
- fine-tuning these thresholds and rigorous quality testing is necessary before the data product can be used. This limits how
- quickly the product can be made available to scientists. Second, the realities of instrument deployment to harsh, remote
- environments mean that instrumentation can go offline periodically, and most conventional algorithms are not able to adapt
- when data inputs are missing. For ARM's thermodynamic cloud phase product, the thermodynamic cloud phase cannot be
- accurately classified when one or more input datastreams are missing.
- Machine learning methods, in combination with conventional methods, can improve thermodynamic cloud phase
- classification. ML algorithms' performance generally improves as they are trained with more data, and they can be trained to
- adapt to data issues such as low quality or missing inputs. There are multiple years of ARM's THERMOCLDPHASE VAP
- data from the NSA site, and the product contains both the VAP and the individual datastreams used to derive it, making it an
- excellent source of training data for the ML algorithms.
- In this work, we develop three machine learning models with increasing complexity: a random forest (RF), a multi-layer
- perceptron (MLP) neural network, and a convolutional neural network (CNN) with a U-Net architecture for classifying
- thermodynamic cloud phase. We use the ARM THERMOCLDPHASE VAP from the NSA site as ground truth for training.
- In addition to evaluation of model performance on NSA data, we evaluate the ML models' generalizability to another ARM
- site (ANX) and test each model's robustness against simulated instrument data loss.

#### 84 **2 Methods**

#### 2.1 Datasets and Data Pre-processing

- This study leverages the THERMOCLDPHASE VAP, produced at the ARM NSA atmospheric observatory
- (https://www.arm.gov/capabilities/science-data-products/vaps/thermocloudphase), as the training data. The ARM NSA site
- (71°19'N, 156°36'W) is located on the northern Alaskan coastline (Verlinde et al., 2016). It experiences a variety of cloud
- types throughout the year, with predominantly ice clouds in winter, mixed-phase clouds in spring and fall, and liquid clouds
- in summer (Shupe, 2011). The observatory is equipped with advanced atmospheric observing instruments, including cloud
- radars, depolarization lidars, radiometers, and radiosondes. These instruments provide comprehensive data for describing cloud
- and radiative processes at high latitudes. These data have been used to improve the representation of high-latitude cloud and
- radiation processes in Earth system models (Balmes et al., 2023; Shupe et al., 2015; Zheng et al., 2023).

The classification algorithm used to create the THERMOCLDPHASE VAP exploits the complementary strengths of cloud radar, depolarization lidar, microwave radiometer, and temperature soundings to classify cloud hydrometeors observed in the vertical column as ice, snow, mixed-phase, liquid, drizzle, rain, and liq driz (liquid+drizzle). The liq driz class represents cases with liquid cloud and drizzle in the same volume, whereas the drizzle class indicates drizzle that has fallen below the cloud. In short, lidar backscatter is sensitive to small cloud droplets with high concentrations, while lidar depolarization ratio can distinguish between spherical (i.e., liquid) and irregularly shaped particles such as ice crystals and snow. Radar reflectivity is dominated by large particles such as ice particles, snow, or raindrops, while higher-order radar moments provide more detailed information on, for example, particle fall speed. Supplemental data, such as liquid water path from the microwave radiometer and temperature profiles from radiosondes, can be used to further refine thermodynamic phase identification. Combining these complementary observations provides a reliable approach to identifying cloud thermodynamic phases. An "unknown" label is assigned in cases when the thermodynamic phase of the hydrometeor cannot be identified due to missing input datasets or when the determined thermodynamic cloud phase is inconsistent with our understanding of cloud structures and physics based on past studies. The VAP also includes a "clear" classification when no hydrometeors are present. A full description of the method is found in Shupe (2007). While Shupe (2007) used lidar and radar measurements to distinguish between clear and cloudy pixels, the THERMOCLDPHASE VAP applies the phase classification algorithm to cloudy pixels identified by the ARM Active Remote Sensing of CLouds (ARSCL) VAP (https://www.arm.gov/data/science-dataproducts/vaps/arscl) (Clothiaux et al., 2001). The ARSCL VAP provides cloud boundaries for up to 10 cloud layers by combining radar, lidar, and radiometer measurements.

The THERMOCLDPHASE VAP reads in micropulse lidar (MPL) or high-spectral-resolution lidar (HSRL) backscatter and depolarization ratio data from the Micropulse Lidar Cloud Mask (MPLCMASK) VAP (https://www.arm.gov/data/science-data-products/vaps/mplcmask) (Flynn et al., 2023) or HSRL data (Goldsmith 2016), respectively; radar reflectivity, mean Doppler velocity, and Doppler spectral width data from the ARSCL VAP; liquid water path data from the Microwave Radiometer Retrievals (MWRRET) VAP (https://www.arm.gov/data/science-data-products/vaps/mwrret) (Gaustad et al., 2011); and temperature data from the Interpolated Sonde (INTERPSONDE) VAP (https://www.arm.gov/data/science-data-products/vaps/interpsonde) (Fairless et al., 2021). The HSRL system is deployed at only a few ARM observatories and ARM Mobile Facility (AMF) field campaigns. When HSRL data are available, the THERMOCLDPHASE VAP uses the HSRL backscatter coefficients and LDR thresholds, as outlined in Shupe (2007), to distinguish between liquid and ice. The MPL system, on the other hand, is deployed at all ARM fixed atmospheric observatories and nearly all AMF field campaigns. The THERMOCLDPHASE VAP uses the gradient of MPL backscatter (MPLGR), following Wang et al. (2001) to distinguish between liquid and ice. We employ the thermodynamic cloud phase classification data that utilizes the MPLGR method to train ML models so that the trained models can be readily extended to other ARM observatories. Ultimately, the THERMOCLDPHASE VAP then outputs seven hydrometeor phase classifications at 30-meter vertical and 30-second

temporal resolutions. The VAP and the datasets used to produce it are publicly available through ARM's Data Discovery tool (<a href="https://adc.arm.gov/discovery/">https://adc.arm.gov/discovery/</a>).

127128129

130131

An example of multi-sensor remote sensing measurements and thermodynamic cloud phase classification from the THERMOCLDPHASE VAP on August 15, 2021, at the ARM NSA observatory is shown in Figure 1. The day started with a deep precipitating system with some embedded convection before 9:00 UTC, with cloud tops reaching up to 8 km and temperatures near the cloud top close to -40 °C. KAZR radar signals can penetrate through the cloud and provide measurements of the cloud structure. Increased radar reflectivity (Z<sub>e</sub>), downward motion (indicated by negative mean Doppler velocity, MDV), and Doppler spectral width (W) around 1 km suggest a transition from cold to warm precipitation (Figure 1c, 1d, and le). Furthermore, the radar bright band is observable at ~ 1km when large falling ice crystals start to become coated with melted liquid water (Figure 1c). Lidar signals, however, were quickly attenuated by warm raindrops near the surface, as shown in Figure 1a and 1b. Large liquid water path (LWP) retrieved from the microwave radiometer and warm temperatures near the surface provide support for this identification (Figure 1f and 1g). As shown in Figure 1g, the THERMOCLDPHASE VAP identifies ice and mixed-phase regions in the middle and upper portions of the cloud system, with snow pixels occasionally present in the middle layers. Below approximately 1 km, warm cloud phases and precipitation, including liquid, drizzle, and rain, are observed. Two additional, relatively shallower cloud systems with similar cloud phase structures were observed between 10:00 and 13:00 UTC and 15:00 and 19:00 UTC. Interestingly, two mid-level thin liquid layer clouds were observed after each of the first two systems. However, due to warmer cloud top temperatures, a lack of ice nucleating particles (INPs), or other processes, these liquid cloud layers did not produce ice or produced ice that was immediately sublimated right below cloud base. Further cloud model simulations could provide insights into these processes (Solomon et al., 2018; Solomon et al., 2011). After 18 UTC, a typical polar low-level stratocumulus cloud with liquid droplets at the top and mixed-phase pixels below is observed. Note that low radar reflectivity that appears to be detached from the cloud below between 20:00 and 22:00 UTC could be artifacts caused by KAZR moderate sensitivity mode (MD) sidelobe impacts (Silber et al., 2018). Accurately detecting and removing these radar artifacts are being investigated by the ARM radar data team (Ya-Chein Feng, personal communication, January 2025).

**Figure 1.** An example of multi-sensor remote sensing measurements of clouds and the thermodynamic cloud phase classification from the THERMOCLDPHASE VAP on August 15th, 2021, at the ARM NSA site. Panels from top to bottom are: a) MPL attenuated backscatter (MPL β); b) MPL linear depolarization ratio (MPL LDR); c) Ka-band ARM zenith radar (KAZR) radar equivalent reflectivity factor ( $Z_e$ ); d) KAZR mean Doppler velocity (MDV); e) KAZR Doppler spectral width (W), f) liquid water path (LWP) from the MWRRET VAP; and g) the thermodynamic cloud phase classification from the THERMOCLDPHASE VAP. Negative MDV values represent downward motions toward the surface. The dashed lines in g) are isothermal lines based on the ARM Interpolated Sonde (INTERPSONDE) VAP.

The various input fields for the VAP are listed in Table 1. These inputs have different units, can differ in scale by orders of magnitude, and may include extreme outlier values. To facilitate training of the neural networks, which can be sensitive to input scaling, each input was range limited and then re-scaled to an approximate range between -2 and 2. The range limiting was chosen to only cut off erroneous or missing datapoints (the ARM datasets assign missing data a value of -9999), and restricts the inputs to a physically plausible range. The scaling values were determined manually based on histograms of the training dataset and are detailed in Table 1. Additionally, the MPL backscatter and MWRRET LWP variables were log-scaled because these observations span several orders of magnitude. Training data were limited to periods where all instruments were operational, and instances where the VAP output was labeled as 'unknown' were excluded from the training process. Based on one year of data from 2021 at the NSA site, 5.9% of cloud hydrometeors were classified as unknown.

Table 1. The formulas used to normalize the input data for the MLP and CNN models. The clip function is used to constrain the values in an array within a specified range E.g.  $clip(x, x_l, x_u) = \{x_l, (x < x_l); x, (x_l \le x \le x_u); x_u, (x > x_u)\}$ 

| Variable                                      | Units            | Lower Bound Upper Bound |      | Full Normalization Formula                  |  |  |
|-----------------------------------------------|------------------|-------------------------|------|---------------------------------------------|--|--|
| MPL backscatter (MPL β)                       | Counts/μs        | 1e-8                    | 1e4  | $(\log(\text{clip}(x, 1e-8, 1e4)) + 6) / 8$ |  |  |
| MPL linear depolarization ratio (MPL Dep)     | NA               | 0                       | 1    | clip(x, 0, 1) * 2 - 1                       |  |  |
| Radar reflectivity (Z <sub>e</sub> )          | dBz              | -70                     | 70   | (clip(x, -70, 70) + 20) / 30                |  |  |
| Radar mean Doppler velocity (MDV)             | m/s              | -8                      | 8    | clip(x + 0.5, -8, 8) / 2                    |  |  |
| Radar spectral width (W)                      | m/s              | -1                      | 4    | clip(x * 5, -1, 4) - 0.5                    |  |  |
| Radar linear depolarization ratio (Radar Dep) | NA               | -20                     | 20   | clip(x + 20, -20, 20) / 6                   |  |  |
| MWRRET liquid water path (LWP)                | g/m <sup>2</sup> | 0.1                     | 2000 | (log(clip(x, 0.1, 2000)) - 3) / 2           |  |  |
| Temperature profile (T)                       | °C               | -100                    | 50   | (clip(x, -100, 50) + 30) / 30               |  |  |

# 2.2 Machine Learning Models

#### 2.2.1 Random Forest

A random forest (RF) is a meta estimator that fits multiple decision tree classifiers using a best-split strategy on various subsamples of the dataset. The individual tree's predictions are then averaged to improve predictive accuracy and control over-fitting (Breiman 2001). The RF model uses an ensemble of 100 decision trees and operates on individual pixels (1.6 million

samples), unlike the CNN, which processes time-height images. We used the Scikit learn library (Pedregosa 2011), to train a random forest classifier, which took less than 2 hours to train using CPUs. The RF was trained using a standard scaler to rescale the input variables and excluding any pixels marked as "unknown" in the VAP. Ninety RF configurations were tested, with the best model determined by considering training accuracy, validation accuracy, and validation F1-score (precision) (Eq. 1). Categorical accuracy evaluates the overall percentage of correct predictions but can be biased in imbalanced datasets. Precision measures the proportion of correct positive predictions out of all positive predictions made. A higher precision indicates fewer false positive predictions. Recall evaluates the proportion of actual positive instances correctly identified. A higher recall indicates fewer false negative predictions. F1-score is the harmonic mean of precision and recall, which is defined as:

$$F1 = 2TP/(2TP + FN + FP)$$
 (1)

F1-score provides a balanced measure of both precision and recall. The best model used 40 trees with 10<sup>5</sup> samples used to train each tree and was trained with a maximum of 2 features used for each split, a maximum depth of 20, and no restriction on the maximum number of leaf nodes. Our initial experiments with the RF model showed that its performance did not change substantially with hyperparameter adjustments, and the best validation performance was achieved using the default scikit-learn hyperparameters.

#### 2.2.2 Neural Network

We also trained a conventional multi-layer perceptron (MLP)-style neural network. The MLP is a supervised learning algorithm that can learn a non-linear, continuous, and differentiable mapping between the input data and the target classifications (Bishop 2006). The MLP takes the 8 normalized input values (Table 1), has 5 hidden layers with ReLU (rectified linear unit) activation functions and 100 neurons each, and a 7-neuron output layer that applies a softmax activation function. Like the RF model, the MLP operates pixel-by-pixel to generate phase classifications. The MLP was also trained using the Scikit-learn library (Pedregosa et al., 2011) with the same dataset used to train the RF, which was standardized as well. Forty-one variants of the MLP were tested with either a robust scalar, quantile transformer, or standard scalar applied directly to the data. The best was trained using the Adam optimizer with an adaptive learning rate initialized at 0.001, a batch size of 200, and a categorical cross entropy loss function. Training was terminated after 134 epochs due to early stopping. The validation fraction was 0.2.

#### 2.2.3 Convolutional Neural Network

Deep convolutional neural networks (CNNs) are powerful machine learning models originally developed for computer vision and image processing tasks (Heaton, 2018; Krizhevsky et al., 2017; LeCun et al., 1998). CNNs learn convolutional kernels that can efficiently represent information about spatial structures in their input fields and are translationally equivariant models, making them optimal for image-recognition and segmentation tasks. Both RFs and CNNs have demonstrated effectiveness in

labeling radar and lidar data for the classification of radial velocity and precipitating hydrometeors (Lu & Kumar, 2019; Veillette et al., 2023). Ronneberger et al. (Ronneberger et al., 2015) introduced the "U-Net," a CNN architecture designed for image segmentation that maps an input image to pixel-level class labels, and several improved, albeit more complex, variants have been developed since its introduction (Huang et al., 2020; Zhou et al., 2018). U-Net and its variants are broadly applicable to both classification- and regression-style image-to-image mapping problems and have now been adapted for a wide range of use cases in the atmospheric sciences (Galea et al., 2024; Geiss & Hardin, 2021; Lagerquist et al., 2023; Sha et al., 2020; Weyn et al., 2021; Wieland et al., 2019). Here, we use a CNN similar to the original U-Net that has been modified for the thermodynamic cloud and precipitation phase retrieval task. The U-Net was implemented using the TensorFlow Keras python library (Chollet 2015) and trained to ingest inputs of size 128x384x8 and produce a 128x384 pixel-level phase classification mask. Missing instrument data values are filled with a value of -9999 prior to dataset normalization, as specified in Table 1. This means that, after normalization, they will be mapped to the lowest allowed value for the corresponding input field by the clip function. On the other hand, if the ground truth labels for any batch of data are missing or classified as unknown, the entire batch is discarded and not used to train the model. The 128x384x8 input shape corresponds to samples that are 1 hour in duration, 12 km in height, and have 8 input fields, respectively. The U-Net was trained using the Adam optimizer with an initial learning rate of 0.001, a batch size of 16, and categorical cross-entropy loss. Training was terminated when mean Intersection Over Union (IOU) reached a maximum value after epoch 10. IOU is defined per-class as:

$$IOU = TP/(TP+FN+FP), \tag{2}$$

Where TP, FN, and FP represent true positives, false negatives, and false positives, respectively. The mean IOU is calculated by averaging the IOU of each class and is not biased by the class imbalance.

**Figure 2.** An illustration of the most effective U-Net architecture tested, showcasing both its encoding and decoding paths along with their channel-dimensions. Given 2-dimensional 8-channel inputs 128x384x8, where the 8 channels are the variables in Table 1 and a cloud mask, the model produces a 128x384x8 output, where each channel represents the softmax probability of a pixel belonging to one of the eight cloud phase classes.

An ablation study was used to determine the optimal U-Net design. This involved altering one design choice at a time, retraining the model, and evaluating the model's performance on the validation data (evaluating on cloudy pixels only, no clear sky). We tested cases with/without dropout layers, channel-wise dropout layers (applied only to the input tensor to simulate instrument dropouts), and batch-normalization layers. We also ran experiments varying the number of convolutional layers in each block, the number of channels in the convolutional layers, the type of activation functions, and the class weights used during training (Ioffe & Szegedy, 2015; Srivastava et al., 2014). During the ablation study, the U-Nets were evaluated using several metrics, including: categorical cross-entropy computed only on cloudy regions, training loss (categorical cross-entropy computed on all regions), mean IOU, and categorical accuracy. The categorical accuracy is averaged over all pixel classifications and, because of class imbalances, is more representative of model skill on the most common classes.

Ultimately, the best U-Net configuration performed the best across all four metrics (lowest cloudy cross-entropy and all-sky cross-entropy and highest mean IOU score and categorical accuracy). The best results with the CNN were achieved with no channel-wise dropout layers, 2 convolutions in each block with the first followed by a dropout layer and the second followed by a batch normalization layer, 64, 64, 64, 128, 128, and 256 channels in the convolutional layers (where the ordering represents depth in the U-Net), leaky ReLU activation functions following the dropout and batch normalization layers, and no class weighting. These design choices resulted in a mean IOU score of 0.810 on the testing dataset, about 0.1 larger than the results of other model configurations we tested. This also resulted in a training loss of 0.018 which was 0.025 less than the other configurations. Notably, the U-Net configuration with channel-wise dropout layers was the second-best model, with an IOU score of 0.528 and training loss of 0.054. We note that these results are based on testing with complete inputs, however, when the U-Net is evaluated with simulated instrument outages the versions that were trained with channel-wise dropout applied to the inputs performed better (details in section 4). The results for all the ablation tests are documented in the Appendix 1.

# 2.3 Training Dataset

Three years of data at the ARM NSA site, from 2018-2020, were used for training and validation and one year of data, from 2021, was used for testing. For the MLP and RF models, a subset of 40,000 pixels from the three years of training data selected randomly were used for each cloud phase, and 10,000 pixels for each cloud phase for validation. For the CNN model, the first 80% of data from 2018-2020 was used for training, and the remaining 20% for validation. The input fields were organized as 3D arrays time x height x channel samples. The 7 unique cloud phase classifications produced by the THERMOCLDPHASE VAP were used as targets (the eighth was clear sky and was not used). The training time for each model is reported in Table

2. The RF and MLP ran on CPUs while the CNN was trained using GPU. For this reason, the MLP and CNN train in comparable time, around ~110 minutes, though the CNN requires more computation. Meanwhile, the RF trains an order of magnitude faster, around 12 minutes. Inference time was inconsequential for all three models, which can each classify a day of data within a few seconds.

The different methods of training set construction and input format used by each of the models creates different class imbalances and inherently complicates a direct comparison between models. For the RF and MLP models, an equal number of samples of each of the cloud phase types were used to train and validate the models because they operate at a pixel level. Meanwhile, the CNN processes full time-height images, and its performance will be biased towards the most common pixel types (ice is the most common class observed at NSA). These challenges inherent to the different ML models, for example, the RF cannot be trained on the huge dataset the CNN uses due to computational constraints. In the future, the CNN could potentially be trained with a class-weighted loss function to ensure the model can identify the minority classes with greater accuracy, but class weighting does not have exactly the same effect as rebalancing the class frequency, particularly when the class imbalance is large. Balancing the class distribution ensures that the model receives gradients of similar scale from each class at approximately the same frequency throughout training. In contrast, altering the class weights results in predominantly small gradients from the majority classes, with occasional large gradients from minority classes. Therefore, achieving optimal performance is likely not as straightforward as selecting class weights that are inversely proportional to class frequency and will likely require fine-tuning of hyperparameters. Recent research has reported better results with combo loss (Taghanaki, 2021) rather than weighting schemes in similar applications (Xie, 2025).

#### 3. Results

- Once the ML models were trained and validated, they were applied to one year of multi-sensor remote sensing measurements from 2021 to predict thermodynamic cloud phase (THERMOCLDPHASE-ML). The predicted phase classifications were
- compared to the VAP to evaluate the performance of the three ML models.

**Figure 3.** Thermodynamic cloud phase classifications from the three ML models and their comparisons against the THERMOCLDPHASE VAP on August 15, 2021, at the NSA site. a)-d) time-height plots of thermodynamic cloud phase classifications from the VAP, as well as from CNN, MLP, and RF model predictions, respectively; e)-g) confidence scores of thermodynamic cloud phase classification predictions from the three ML models; h)-k) histograms of thermodynamic cloud phase distributions; l)-n) normalized confusion matrices for each model. Figure 3a is identical to Figure 1g.

THERMOCLDPHASE VAP on August 15, 2021, at the ARM NSA site. Among the predictions from the three ML models. the CNN demonstrates the best agreement with the THERMOCLDPHASE VAP, capturing nearly identical thermodynamic cloud phase structures. The MLP and RF models also show good agreement with the VAP but tend to overestimate mixedphase pixels in the ice-dominated high clouds between 0:00 –9:00 and 15:00 – 18:00 UTC and underestimate ice-phase pixels in the low-level clouds 15:00 – 23:00. Notably, the ML models provide confidence scores for their predictions, where higher scores indicate greater certainty. For the CNN and MLP models the raw model output is a softmax probability score for each phase class. For the RF confidence is calculated using the mean of predictions of trees in the RF. As shown in Figures 3e–3g, the CNN consistently generates higher confidence scores compared to the MLP and RF models. Regions with low confidence scores from the MLP and RF models often correspond to areas where these models misclassify thermodynamic cloud phases. As shown in Figures 3e-g, all ML models exhibit significantly lower confidence scores within the melting layer—a region characterized by rapid transitions in particle phase, shape, and fall speed. While this zone is critical for understanding cloud regime shifts, it remains difficult to resolve. Improving detection in this region will require a refined training dataset specifically focused on the melting layer, which remains an active area of research (Brast and Markmann, 2020; Xie et al., 2024). In supplementary Figure S1 we plot confidence score bins versus accurate classifications for the 2021 data. The MLP and RF models' accuracy linearly increases with higher confidence. For the CNN, for confidence scores above 40% accuracy linearly increases. There is a local maximum in accuracy for low confidence scores between 20-30%. For these cases, there are several orders of magnitude fewer data points, and the majority of correctly classified cases are ice. Because NSA is dominated by ice, classifying a non-clear-sky pixel as ice, even with low confidence, has a high chance of being correct for this dataset. At the pixel level of the thermodynamic cloud phase classification, Figure 3h indicates that the day was dominated (volume-wise) by ice-phase pixels, followed by liquid and mixed-phase pixels. Small amounts of snow, drizzle, and liq driz pixels were also identified. The histogram plots of ML-predicted thermodynamic cloud phases in Figures 3i–3k show that the CNN produces a histogram closely matching the VAP. In contrast, the MLP and RF models tend to underestimate ice-phase pixels while overestimating liquid, mixed-phase, and liq driz pixels. Figure 31–3n provides a more detailed evaluation of thermodynamic cloud phase classifications from the three models through confusion matrices. The multi-class confusion matrix is a 7x7 grid with a row and column for each of the cloud phases. Each row represents the class reported by the VAP, and columns show the class predicted by ML. Correct predictions (true positives)

Figure 3 provides an example of thermodynamic cloud phase classifications from the three ML models compared with the

Figure 31–3n provides a more detailed evaluation of thermodynamic cloud phase classifications from the three models through confusion matrices. The multi-class confusion matrix is a 7x7 grid with a row and column for each of the cloud phases. Each row represents the class reported by the VAP, and columns show the class predicted by ML. Correct predictions (true positives) are found along the diagonal, while misclassifications are in the off-diagonal elements. The sum of the columns ideally would equal one; a sum greater than one indicates an over classification of pixel type. On this day, liquid, mixed-phase, drizzle, and snow pixels were accurately identified by all three ML models, with accuracies exceeding 0.8. While the CNN model also accurately classified ice-phase pixels, the MLP and RF models frequently misclassified them as liquid or mixed-phase pixels. This case has pixel percentages above 5% for all cloud phase types and has high accuracy for all types, liq\_driz and rain pixels included. In cases consisting of predominately ice clouds, relatively low accuracy for liq\_driz and rain pixels are reported

compared to other categories, with the CNN performing the worst, likely due to the extremely low occurrence of these pixel types and an overzealousness for predicting ice.

#### 3.2 Analyses of ML Model Performance

Given that the confidence score reflects the uncertainty of ML predictions, it is essential to analyse confidence scores and their relationship to accuracy for different thermodynamic cloud phases. Figure 4 presents a comprehensive statistical analysis of ML model confidence scores based on one year of data from 2021 at the NSA site. Overall, the confidence scores for thermodynamic cloud phase predictions peak near 100%, which is promising. Among the phases, predictions for ice, rain, and snow generally exhibit higher confidence scores across all three ML models. The ice phase, in particular, is reliably predicted especially by the CNN model—due to the availability of key information such as lidar backscatter and depolarization ratio, radar reflectivity, mean Doppler velocity and spectral width, and temperature (Shupe, 2007). The rain and snow phases, representing large particles in warm and cold conditions respectively, can be identified using key information such as radar reflectivity, mean Doppler velocity, and temperature. In contrast, the confidence scores for the "liquid" phase predictions are lower than those for the ice, rain, and snow phases. While the liquid phase can theoretically be reliably determined using lidar backscatter and depolarization ratio measurements, lidar signals are often quickly attenuated by low-level clouds, as illustrated in Figures 1a and 1b. Under such conditions, identifying liquid-phase pixels becomes challenging when relying solely on radar reflectivity and spectral width data (Silber et al., 2020). The mixed, drizzle, and liq driz phases have even lower confidence scores, likely due to the inherent difficulties in extracting their distinguishing characteristics from available measurements. Among the three ML models, the CNN achieves the highest confidence scores across all thermodynamic cloud phases. The MLP model exhibits confidence scores comparable to the RF model for liquid, ice, mixed-phase, drizzle, and liq driz phases but shows significantly lower confidence scores for the rain and snow phases.

**Figure 4:** Probability Density Functions (PDFs) of confidence scores for thermodynamic cloud phase predictions from the three ML models using one year of data in 2021 at the NSA site.

Figure 5 shows the frequency of thermodynamic cloud phases at the NSA site as derived from the THERMOCLDPHASE VAP (labelled as 'VAP') and predictions from the three ML models using one year of testing data. Due primarily to the low polar temperatures, the NSA site is primarily dominated by the ice phase, followed by mixed, snow, and liquid phases. Warm phases, including drizzle, liq\_driz, and rain, occur much less frequently and are mostly confined to the summer season (Shupe, 2011). Comparing the ML predictions with the THERMOCLDPHASE VAP, the CNN closely matches the VAP's percentage distribution of thermodynamic cloud phases. In contrast, both the MLP and RF models predict lower percentages for the ice phase but higher percentages for the liquid, mixed, drizzle, and liq\_driz phases, consistent with the case observed in Figures 3h–3k.

**Figure 5.** Percentage distributions of thermodynamic cloud phases from the THERMOCLDPHASE VAP (labeled as 'VAP') and predictions from the three ML models, based on one year of data from 2021 at the NSA site.

Figure 6 presents the confusion matrices for the three models computed on the testing set. All models achieved over 80% accuracy for each cloud phase. The correct prediction percentages are close for the three ML models except that CNN has dramatically higher correct prediction for ice than the other two ML models. The CNN correctly identified ice 99% of the time. However, it occasionally misclassified liquid (8%), mixed (12%), and drizzle (1%) as ice. Because there are so few total instances of these phases (Figure 5), these misidentifications did not contribute much to reducing the overall accuracy of the model. However, to do a true comparison of the models to the best of our ability, we retrained the RF and MLP models on a random sample of 1.6 million pixels from the training dataset (using the same number of samples as the class-balanced training and same the inputs and normalizations used by the CNN) where the distributions of phases match closely with the overall

phase distribution in the VAP. We examined how the "imbalanced" RF and MLP compared to the CNN (Figure S2). Focusing on the prediction of ice, the "balanced" RF and MLP models only misclassify liquid and mixed phase as ice 4% and 5% of the time, respectively (Figure 6), while the "imbalanced" RF misclassifies liquid and mixed phase 25% and 22% of the time and the "imbalanced" MLP misclassifies them 22% and 21% of the time (Figure S2). Regarding the performance of the "imbalanced" models on the warm cloud phases, for drizzle, the CNN correctly identifies it 83% of the time, the imbalanced RF 86%, and the imbalanced MLP 81%. Compared to the balanced RF (90%) and MLP (88%), the imbalanced datasets perform worse on this metric.

**Figure 6.** Confusion matrices computed on the 2021 NSA test dataset for (a) the CNN U-net model, (b) the MLP model, and (c) the RF model. The values are normalized by row, with the main diagonal showing true positive predictions and values off the main diagonal representing incorrect predictions.

The performance of the three ML models was statistically evaluated using performance metrics listed in Table 2. These metrics include categorical accuracy, precision, recall, F1-score, and mean IOU (Eq.1). Here, we calculated the test accuracy as the percentage of pixels that match the VAP. Precision, recall, F1-score, and IOU are calculated for each phase class and reported as an average across the classes to reduce bias due to class imbalance. These metrics provide us with information to evaluate the performance of the three ML models in classifying thermodynamic cloud phases on a pixel-by-pixel level.

Table 2. Model performance metrics for the three machine learning models on the test dataset.

| Model | Accuracy (%) | Precision* | Recall* | f1-score* | IoU*  |
|-------|--------------|------------|---------|-----------|-------|
| CNN   | 95.7         | 0.890      | 0.894   | 0.891     | 0.811 |
| MLP   | 85.7         | 0.760      | 0.905   | 0.815     | 0.704 |

| RF 87.2 0.789 0.913 0.837 0.735 |
|---------------------------------|
|---------------------------------|

\*using a macro average across classes

Table 2 shows that each model agreed with the THERMOCLDPHASE VAP in more than 85% of the utilized samples. The CNN achieved the highest test accuracy, F1-score, and mean IOU. The RF model performed slightly better than the MLP across these metrics but was significantly outperformed by the CNN. We hypothesize that the CNN's superior performance is due to its ability to evaluate the input time-height arrays (sections of data covering 11km in height by 1 hour) holistically rather than on a pixel-by-pixel basis. This approach allows the CNN to leverage information from neighbouring pixels and potentially assess larger-scale features, such as cloud shape, to improve classification accuracy.

**Figure 7.** Vertically resolved ML model F1-scores and mean IOU scores, overlaid on a stacked histogram of the frequency of the cloud thermodynamic phase categories. A height bin size of 0.5 km is used to calculate the vertical profiles of mean IOU and F1-scores. Noise around 7.5-10km likely due to phase extinction and low pixel count.

Another aspect of evaluation is the performance of the models with respect to altitude. Figure 7 presents vertically resolved F1-scores and mean IOU scores for the ML models, overlaid on a stacked histogram of cloud thermodynamic phase category occurrences based on the VAP. Vertically resolved cloud phases converge toward ice-only clouds due to the extremely cold environment at higher altitudes. A peak in liquid phase occurrence is observed around ~1 km, which may be due to the prevalence of low-level polar stratiform mixed-phase clouds with a thin liquid layer at the top in the VAP (Silber et al., 2021; Zhang et al., 2010; Zhang et al., 2017), or due to the artifacts caused by KAZR MD mode (MD) sidelobe as discussed in section 2.1. The F1-scores and mean IOU are consistent with altitude until 8km when they start to increase across the three ML models, primarily due to the higher frequency of the ice phase at greater altitudes and the fact that the ice phase is more reliably predicted by all three ML models, as shown in Figure 4b. The CNN consistently achieves significantly higher F1-scores than the MLP and RF models at altitudes below ~6 km. This is attributed to the greater diversity of thermodynamic cloud phases at lower altitudes and the CNN's strong performance across all phases, as shown in Figure 4.

### 3.3 Input Feature Importance

To identify which input features are most influential in determining cloud phase and to provide additional context for model performance, we calculate permutation feature importances (Breiman, 2001) for the three ML models by cloud phase class. We assess the permutation importance of an input feature defined as the model's recall score for a specific phase category on the test set minus its recall score resulting from shuffling the values of the input feature (randomly reordering their positions within the column), which effectively removes its relationship with a specific phase category. A significant difference between recall scores indicates that the feature is important, while little or no change suggests the feature has minimal importance. This is done for each phase class, and the recall score is used specifically because it shows the reduction in the models' ability to positively identify specific thermodynamic phases. This process is repeated for the CNN, MLP, and RF models, and is reported in Figure 8.

Overall, input features from radar measurements (panels b, f, and j) including  $Z_e$ , MDV, and W, and radiosonde temperature measurements (panels d, h, and l) are the most significant for classifying thermodynamic cloud phases across all the three models. In contrast, input features from lidar measurements (panels a, e, and i) and the MWRRET LWP (panels c, g, and k) are less influential, probably because lidar signals are quickly attenuated by persistent low-level clouds at the NSA site (Shupe et al., 2011; Zhang et al., 2017), and LWP provides only column-integrated information rather than detailed vertical profiles. Future work may want to explore the feature importance restricted to pixels that were observed by both radar and lidar to reevaluate the lidar importance.

**Figure 8.** Permutation feature importances of predicting thermodynamic cloud phases from: (a-d) the CNN Unet model; (e-h) the MLP model; and (i-l) the RF model. Features are grouped by the instruments they are derived from. The abbreviations are defined in Figure 1. Radar Dep represents radar linear depolarization.

Colors in Figure 8 represent different phase categories and enable feature importance to be assessed for each category. The main focus of permutation feature importance is the relative importance of the features instead of their absolute values. This is because the sum of the importances is not necessarily meaningful, given that feature interactions and the non-additive nature of the method can affect the results. For the CNN model: radar Z<sub>e</sub>, MDV, and MPL β are identified as the three most important input features for determining the liquid phase. This aligns with the logic used in threshold-based algorithms by Shupe (Shupe, 2007) for liquid phase identification. As shown in Figure 8a, lidar backscatter shows notable importance in the CNN model. While lidar backscatter and depolarization ratio offer direct and reliable indicators of liquid-phase presence, radar-based variables—such as reflectivity, mean Doppler velocity, and spectral width—can also contain useful signatures of liquid-phase clouds (Luke et al., 2010; Yu et al., 2014; Kalogeras et al., 2021; Schimmel et al., 2022), as evidenced in Figure 8b, f, and j. Lidar measurement's lower feature importance in deference to radar measurements was also observed on days with single-layer, low-level liquid clouds (Figure S3). For the ice phase, input feature importances are generally lower, likely because the

ice phase can be independently identified using multiple input features. As a result, even when one input feature is missing, the ice phase can still be accurately classified using the remaining features. The key features for identifying the mixed-phase are Z<sub>e</sub>, MDV, and W. For drizzle, liq\_driz, and rain, Z<sub>e</sub>, MDV, and temperature are most important, likely due to the complexity of distinguishing these phases, requiring multiple measurements. Z<sub>e</sub> is the primary feature for snow identification, followed by MDV and temperature, consistent with Shupe (Shupe et al., 2015), where snow identification relied on Z<sub>e</sub> and temperature. The importance of MDV for snow may result from its covariance with Z<sub>e</sub>. The input feature importances for the other two models (Figures 8b and 8c) are generally similar to those of the CNN model. Broadly, the feature importances in Figure 8 align with physical intuition and with the logic used by Shupe, (2007), indicating that ML models successfully captured the relationships between remote sensing measurements and the thermodynamic cloud phases.

#### 3.4 Application to another ARM Site: COMBLE

To assess the generalization capability of the ML models, we applied them at a different ARM Mobile Facility (AMF) observatory. The ARM Cold-Air Outbreaks in the Marine Boundary Layer Experiment (COMBLE) field campaign deployed an AMF at a coastal site in Andenes, Norway (69.141° N, 15.684° E, referred as the 'ANX' site) from December 2019 to May 2020 (Geerts et al., 2022). The campaign aimed to investigate the relationships between surface fluxes, boundary layer structure, aerosol properties, cloud and precipitation characteristics, and mesoscale circulations during cold-air outbreaks (CAOs) over open Arctic waters (Geerts et al., 2022). A key focus was to enhance understanding of thermodynamic cloud phases and their evolution during CAOs. The deployment at the main site included all remote sensing measurements required to run the THERMOCLDPHASE VAP, as well as the input features needed for the ML models. However, MPL data was missing until February 11, 2020. Consequently, the THERMOCLDPHASE VAP between February 11 and May 31, 2020, was produced for this site shortly after the field campaign and has since been utilized in recent studies to analyse cloud phase structures over the polar regions (Lackner et al., 2024; Van Weverberg et al., 2023; Xia & McFarquhar, 2024).

**Figure 9.** Figure 9 has the same subplot structure as Figure 3. Figure 9 shows the results of applying the ML models to data collected at the ANX site in Norway on February 25th, 2020, during the COMBLE campaign. The case day chosen is experiencing a CAO event. Note at 00:00 UTC in subplot (a) the VAP has unknown pixels, which the ML models are able to resolve (b, c, d).

We evaluated the models' ability to classify thermodynamic cloud phases for a CAO event identified on February 25th, 2020. Figure 9 presents thermodynamic cloud phase classifications from the THERMOCLDPHASE VAP, the three ML model predictions, and evaluations of ML model performance against the THERMOCLDPHASE VAP. Convective cloud structures

and production of heavy snowfall during the CAO can be clearly observed from the time-height plot of thermodynamic cloud phases in Figure 9a. ML model predictions compare well with the THERMOCLDPHASE VAP (Figure 9b-9d). A figure with the data streams used to create the VAP similar to Figure 1 is available in the supplement (Figure S4). All three models captured the time period accurately, with ice and snow dominating the ML-classified thermodynamic cloud phases. Interestingly, there are some "unknown" phase pixels at the beginning of the day from the THERMOCLDPHASE VAP where the static algorithm was unable to resolve the cloud phase because the phase identification is inconsistent with our understanding of cloud physics based on past studies. Large Z<sub>e</sub> and cold temperatures suggest these pixels are snow, yet they exhibit falling velocities exceeding 2.5 m/s. Snow typically has low terminal velocities due to its small mass density and large surface area. However, during the CAO event's strong convective conditions, snow velocities may increase significantly in intense downdraft regions. The three ML models consistently predicted "snow" in this region, which is consistent with surrounding pixels, demonstrating an advantage of using ML models for cloud phase classifications.

Both the CNN and MLP have high confidence scores that are generally greater than 90% for ice and snow pixels but significantly lower confidence scores for liquid and mixed-phase pixels. Indeed, it is challenging to reliably distinguish liquid and mixed-phase pixels from ice phase pixels when they are embedded in ice-dominated clouds (Shupe, 2007; Silber et al., 2021). The lower model performance at ANX compared to NSA is likely due to the more complex convective cloud structures associated with cold-air outbreaks (CAOs) at ANX (Geerts et al., 2022). The RF has lower confidence scores except for ice phase pixels at high altitudes after 12:00 UTC. The histogram plots in Figures 9i–9k show that all three ML models produce histograms that closely match the VAP, with the MLP and RF models slightly over-predicting the "liquid" category and underpredicting the "ice" category. The confusion matrices in Figure 9l–9n confirm that all three ML models predict the dominant ice and snow phases reasonably well, with accuracies exceeding 0.85. The three models all showed lower accuracy for the liquid phase (<0.7), which is a minority category in this sample. In addition, both the MLP and RF showed good predictions of the mixed-phase pixels, while the CNN showed a much lower accuracy in predicting mixed-phase pixels for this day. Overall, the CNN outperformed the MLP and RF models in terms of accuracy when predicting the dominant categories but performed worse than the other two models when predicting the minority categories.

Model performance metrics for the entire study period in which the THERMOCLDPHASE VAP was produced at ANX are reported in Table 3. ANX plots in the same format as those produced for NSA (Figures 4, 5, and 6) are presented in supplemental Figures S5, S6, and S7. Every performance metric using ANX as a test dataset (accuracy, precision, recall, F1-score, and IOU) is reduced in comparison to NSA (Table 3). The NSA test dataset comprised 12 months of data, and the ANX dataset comprised 4 months of data (February - May). Comparing the PDFs of confidence scores for the cloud phase predictions for the three models, differences emerge. The CNN model behaved similarly at both sites, likely because the CNN incorporates information from neighbouring pixels and because of the prevalence of ice at both locations, and for all phases predictions peaked at 100% confidence (Figure S5). The RF model also peaks at 100% for all phases except for liquid and liq\_driz peaking at 90% and displaying a secondary local maximum at 40 percent. The MLP diverges the most with only the PDF of ice

classification confidence peaking at 100%. The PDFs for all other phases for the MLP model are more symmetrical and peak between 50 - 60%. In addition, all three models reported higher false negatives for drizzle, liq\_driz, and rain (Figure S7). Comparing frequency distributions of cloud phases, ANX and NSA are similar as they are both high-latitude locations. Ice accounts for ~60% of all cloud phases detected, followed by mixed, snow, and liquid (Figure S6).

Table 3. Model performance metrics for the three ML models on the dataset from COMBLE at ANX

| Model | Accuracy (%) | Precision* | Recall* | F1-score* | IOU*  |
|-------|--------------|------------|---------|-----------|-------|
| CNN   | 92.5         | 0.841      | 0.777   | 0.805     | 0.69  |
| MLP   | 80.4         | 0.684      | 0.827   | 0.725     | 0.594 |
| RF    | 81.1         | 0.703      | 0.806   | 0.726     | 0.597 |

<sup>\*</sup>using a macro average for each output class

#### 4. Data Dropout Experiment (Improving Threshold Algorithm)

One advantage of using machine learning models for thermodynamic phase classification is that, unlike the VAP, they can still provide classifications in missing data scenarios. To assess model robustness against missing inputs, we tested our models by systematically removing either a single variable or all variables from a specific instrument to simulate scenarios where the instrument was offline. We also trained a variant of the U-Net designed to be resilient to missing data by including a layer to drop-out random input channels with a likelihood of p=0.125 during training, referred to as "CNN-ICD" (input channel dropouts). The CNN-ICD model was the second-best performing CNN in the ablation study in Section 2.2.3, when all input channels were used, but the addition of the input channel dropout during training makes it far more robust in missing data scenarios.

We tested our models on a year's worth of data in 2021 at the NSA site. For each test, we evaluated IOU score for each cloud phase type over the year, the overall mean (with respect to phases) IOU score, and the total accuracy. Table 4 shows the results for the CNN-ICD model. Results for the other models are in the supplement (Tables S2-4, Figure S8). The two instruments that had the greatest effect on accuracy were the radiosonde temperature and the radar datastreams. For instance, for 2021, the accuracy of the CNN dropped from 95 to 88% without temperature data (mean IOU dropped 0.81 to 0.37), and the accuracy of the RF dropped from 86 to 74% (IOU 0.72 to 0.28) (Table S2 and Table S4). The CNN-ICD model in comparison with temperature dropped from 88% to 85% accuracy and 0.62 to 0.55 IOU, so while its control case performs worse, it is lease affected by data outages. It is also worthwhile to note how and where the cloud phase classification failed without certain instruments. Dropping the MWR data had minimal effect on model performance for all four models. However, without the radar mean doppler velocity, the CNN, for example, had trouble distinguishing rain and drizzle in liquid clouds. This is because Doppler velocity is key for determining whether a liquid particle is falling (Shupe 2007). Another example is temperature, without which the model has trouble distinguishing solid from liquid water phases.

**Table 4.** Performance of the CNN-ICD model in the data dropout study.

| CNN IC  | Intersection over Union (IOU) Score                 |        |       |       |         |                   |       |       |          |                     |
|---------|-----------------------------------------------------|--------|-------|-------|---------|-------------------|-------|-------|----------|---------------------|
| Model   | Missing<br>datastream/<br>Instrument                | Liquid | Ice   | Mixed | Drizzle | Liquid<br>Drizzle | Rain  | Snow  | Mean IOU | Total<br>Accuracy % |
| CNN-ICD | Control                                             | 0.441  | 0.875 | 0.530 | 0.426   | 0.429             | 0.849 | 0.808 | 0.622    | 88.4                |
| CNN-ICD | Micropulse Lidar,<br>all datastreams                | 0.535  | 0.894 | 0.555 | 0.412   | 0.546             | 0.844 | 0.890 | 0.668    | 90.2                |
| CNN-ICD | Micropulse Lidar,<br>backscatter                    | 0.467  | 0.877 | 0.553 | 0.362   | 0.463             | 0.850 | 0.860 | 0.633    | 88.7                |
| CNN-ICD | Micropulse Lidar,<br>linear<br>depolarization ratio | 0.469  | 0.877 | 0.508 | 0.407   | 0.448             | 0.841 | 0.819 | 0.624    | 88.6                |
| CNN-ICD | Microwave<br>Radiometer                             | 0.436  | 0.876 | 0.533 | 0.438   | 0.440             | 0.850 | 0.802 | 0.625    | 88.5                |
| CNN-ICD | Radar, all<br>datastreams                           | 0.180  | 0.800 | 0.001 | 0.103   | 0.244             | 0.003 | 0.204 | 0.219    | 76.8                |
| CNN-ICD | Radar, linear<br>depolarization ratio               | 0.432  | 0.869 | 0.525 | 0.388   | 0.411             | 0.849 | 0.799 | 0.611    | 87.9                |
| CNN-ICD | Radar, mean doppler velocity                        | 0.347  | 0.891 | 0.374 | 0.488   | 0.467             | 0.694 | 0.836 | 0.585    | 89.2                |
| CNN-ICD | Radar, reflectivity                                 | 0.374  | 0.870 | 0.445 | 0.450   | 0.500             | 0.770 | 0.109 | 0.502    | 84.3                |
| CNN-ICD | Radar, spectral width                               | 0.459  | 0.879 | 0.470 | 0.600   | 0.316             | 0.802 | 0.873 | 0.629    | 88.9                |
| CNN-ICD | Radiosonde<br>Temperature                           | 0.143  | 0.883 | 0.367 | 0.450   | 0.456             | 0.788 | 0.809 | 0.557    | 88.5                |

Table 4 shows that the CNN-ICD model performs well even with missing data, generally achieving a mean IOU > 0.5 and accuracy > 75%. We hypothesise that with the addition of the 2D dropout layers, which mimic instrument dropouts, it had greater elasticity to adapt to missing data and thus will be more robust to these events. When all input fields are available, we achieved the best results without the addition of these layers however. Interestingly, in some cases the CNN-ICD model has greater accuracy and IOU score if some of the instrument datastreams are missing, such as the linear depolarization ratio for the lidar and radar. This could indicate that some of the datastreams give conflicting phase information or add input noise, in which case their inclusion actually makes the model less robust. We do not see this with the other models however.

**Figure 10:** Example of how each model responds to missing temperature data from the interpolated sonde on August 15, 2021, at the NSA site. (a–e) time-height plots of thermodynamic cloud phases from the THERMOCLDPHASE VAP (Ground Truth VAP), CNN-ICD, CNN, MLP, and RF, respectively; (f–i) thermodynamic cloud phase classifications from the four ML models when temperature data is dropped out from the input features; (j–m) the differences in thermodynamic cloud phase classifications between model predictions with and without temperature data for the four ML models.

Figure 10 demonstrates how each model responds to the absence of temperature data from the interpolated sonde on August 15, 2021, at the NSA site. This temperature data was identified as one of the most important input features for all the ML models in Figure 8. On this day, deep clouds were observed at the beginning and end of the day and low-level clouds during the middle of the day. Due to elevated surface temperatures, the low-altitude clouds were predominantly composed of warm classes. This case serves as an excellent example for the data dropout experiment, as it includes all thermodynamic cloud phases. When all input features are available, the four ML models demonstrate strong performance compared to the THERMOCLDPHASE VAP (Figures 10a-e). When temperature data are removed, all models show reduced performance (Figures 10f-i), with the "CNN-ICD" model exhibiting the smallest reduction in performance. It accurately identifies mid- and high-level cloud phases but misclassifies liquid, drizzle, and rain as ice, mixed-phase, and snow, particularly for low-altitude cloud pixels at the beginning and end of the day when temperature data are missing (Figure 10f). Interestingly, the "CNN-ICD" model still correctly identifies low-altitude warm cloud classes between 3:00 and 20:00 UTC. The CNN, MLP, and RF models also correctly classify thermodynamic cloud phases for mid- and high-level cloud pixels but frequently misclassify

liquid, drizzle, and rain as ice, mixed-phase, and snow for low-altitude cloud pixels throughout the day (Figures 10g and 10h). The responses of each ML model to the removal of other input features are detailed in different rows in Figures S9 and S10. Overall, the CNN-ICD model performs the best in the absence of data, followed by the CNN and then the MLP and RF models perform roughly equally.

**Figure 11.** An example of how the CNN trained with input channel dropouts (CNN-ICD) responds to different missing input variables, mimicking data loss in the field for the same case shown in Figure 10. The title of each panel shows the variable or all variables from a specific instrument that was dropped out. 'All MPL Var' and 'All Radar Var' represent all lidar variables and all radar variables were dropped out, respectively.

Figure 11 shows how the CNN-ICD model responds to the removal of different variables for predicting thermodynamic cloud phases for the same case shown in Figure 10. Consistent with the input feature analysis shown in Figure 8, removing the MPL β, MPL Dep, radar Dep, LWP, and all MPL variables has minimal impact on the performance of the CNN-ICD model. When Ze is missing, the model sometimes fails to distinguish between liquid and drizzle for low-altitude cloudy pixels throughout the day and between ice and snow for mid- and high-level cloud pixels at the end of the day (Figure 11c). Without radar W, the model sometimes fails to identify mixed-phase pixels for mid-level clouds, although they are only present for short periods in this example (Figure 11e). Dropping out radar MDV causes the model to sometimes fail to distinguish between rain and drizzle between 3:00 and 6:00 UTC (Figure 11f). Dropping out T causes the model to sometimes fail to distinguish between ice and liquid at the beginning of the day and between ice and drizzle at the end of the day (Figure 11g). Overall, dropping out individual radar variables (including Ze, MDV, W), all radar variables simultaneously, or dropping out temperature data had the largest effect on predicting thermodynamic cloud phases. This general result is also true for the other ML models for this

case study, which are detailed in Figures S9 and S10. This result shows general agreement with the feature importance results presented in Section 3.3.

# 5. Summary and Conclusions

- The ARM THERMOCLDPHASE VAP offers vertically resolved cloud thermodynamic phase classifications using the multi-578 sensor approach developed by Shupe (2007), which combines lidar backscatter and depolarization ratio, radar reflectivity. 579 Doppler velocity and spectra width, liquid water path, and temperature measurements. This study leveraged multiple years of 580 the VAP product as the ground truth to train and evaluate three ML models for identifying thermodynamic cloud phases based 581 on multi-sensor remote sensing data collected at the ARM NSA observatory. The models are a RF, a MLP, and a CNN with a 582 U-Net architecture. Input features for the three ML models include MPL β and MPL Dep, radar Z<sub>e</sub>, MDV, W, and Radar Dep, 583 MWR derived LWP, and radiosonde T. An ablation study was conducted to find the optimal configuration of the CNN model. 584 Three years of data at the ARM NSA site, from 2018-2020, were used for training and validation, while one year of data, from 585 2021, was used for testing. The input fields were organized as 3D arrays (time x height x channel), with the channel dimension 586 containing the nine individual ARM datastream inputs. The seven unique cloud phase classifications produced by the 587 THERMOCLDPHASE VAP were used as target variables.
- The three trained ML models were applied to one year of multi-sensor remote sensing measurements from 2021 to predict 589 thermodynamic cloud phase (THERMOCLDPHASE-ML). The accuracy of these predictions was evaluated against the outputs 590 of the THERMOCLDPHASE VAP. Evaluations included a detailed one day case study and year-long statistical assessment 591 using performance metrics such as categorical accuracy, precision, recall, F1-score, and mean IOU. Among the ML models, 592 the CNN demonstrated superior performance, achieving the highest categorical accuracy, F1-score, and mean IOU. This 593 success is likely attributed to its holistic evaluation of input time-height arrays rather than the pixel-by-pixel approach used by 594 the MLP and RF models. The CNN's success may also be due to site dependency, as NSA is ice dominated, and this model 595 best predicts ice. The evaluations were further extended to data from an ARM AMF observatory during the ARM Cold-Air 596 Outbreaks in the Marine Boundary Layer Experiment (COMBLE) field campaign at a coastal site in Andenes, Norway.
  - We also demonstrated three possible advantages of using ML models for thermodynamic cloud phase classification, including:
    - 1) ML models provide confidence scores for their predictions, with higher scores indicating greater certainty. Statistical analysis of one year of ML classification data reveals that predictions for ice, rain, and snow generally exhibit higher confidence scores, followed by the liquid phase. The mixed, drizzle, and liq\_driz phases show lower confidence scores. Among the three ML models, the CNN produced the highest confidence scores across all thermodynamic cloud phases.
    - 2) ML models enable feature importance analysis to identify the input features most influential in determining thermodynamic cloud phases. Analyzing the calculated permutation feature importances for the three ML models

reveals that radar moments - specifically Z<sub>e</sub>, MDV, and W - as well as temperature, are the most significant features for classifying thermodynamic cloud phases. In contrast, input features from lidar measurements and MWRRET LWP were found to be less influential.

618

629

3) ML models can predict thermodynamic cloud phases even when one or more input data set is missing. To evaluate this capability, we conducted data dropout experiments by systematically removing either a single input variables or all variables from a specific instrument to simulate scenarios where the instrument was offline. We also trained a CNN U-NET model with input channel dropouts during training (referred to as " CNN-ICD "), hypothesizing that the inclusion of channel-wise dropout layers would mimic real instrument dropouts and enhance the model's ability to adapt to missing data, thus making the model more robust. Overall, the CNN-ICD model performs better than the others when input fields are missing, followed by the standard CNN and MLP models, with the RF model performing the worst. Dropping out radar variables, including radar Z<sub>e</sub>, MDV, W, and all of them together, as well as dropping out temperature data, had the largest negative impacts on predicting thermodynamic cloud phases for all models.

We utilized thermodynamic cloud phase classifications from the THERMOCLDPHASE VAP as the ground truth. However, the VAP, which employs empirical threshold-based algorithms, can misclassify thermodynamic cloud phases (Shupe 2007). Therefore, we do not expect the trained ML models to produce better thermodynamic cloud phase classifications than the THERMOCLDPHASE VAP in most cases. Instead, we demonstrated the feasibility of using ML models to predict thermodynamic cloud phase classifications with accuracy close to the VAP while adding additional information, such as confidence scores and feature importances. Furthermore, ML models can extend classification to scenarios where some instruments are offline, which are typically problematic for the VAP, and can produce reasonable classifications in some specific cases when the VAP algorithm cannot. The ML models demonstrate elasticity in their ability to classify cloud phase. such as when the VAP was unable to classify snow in the COMBLE case study. Even so, we note that CNNs have limited interpretability and are less physics-informed than a hand-crafted retrieval. There are other more advanced segmentation algorithms than U-Nets that could be tested in future studies, e.g., U-Net++ (Zhou et al., 2018; King et al., 2024) and vision transformers (Springenberg et al., 2023; Thisanke et al., 2023; Lenhardt et al., 2024). Furthermore, feature or saliency map analysis could offer valuable insights into whether the CNN focuses on physically meaningful regions of the data, and represents a promising direction for future work to enhance the interpretability of model predictions and aid future model development (Haar et al., 2023). Our next step will involve creating a multiple-year, expert-labeled dataset of thermodynamic cloud phases to train ML models. The goal is to have a ML model that ultimately predicts better thermodynamic cloud phases than those derived from empirical threshold-based algorithms. It is important to note that the definition of thermodynamic phases depends on instrument sample volume and detection limit (Korolev & Milbrandt, 2022). The seven thermodynamic cloud phase categories used in this study are empirical and might not precisely represent true thermodynamic cloud phases in nature. Therefore, we also plan to explore using unsupervised machine learning schemes for classifying thermodynamic cloud phases, using the THERMOCLDPHASE data only as a reference of comparison in future work.

Environmental Research (BER) program. This research was supported by the DOE ARM program. M.D.S. was supported by

the DOE (DE-SC0021341), NOAA Cooperative agreement (NA22OAR4320151), and the NOAA Global Ocean Monitoring

and Observing Program (FundRef https://doi.org/10.13039/100018302).

665 666

672

681

684

691

- Adebiyi, A. A., Zuidema, P., Chang, I., Burton, S. P., & Cairns, B. (2020). Mid-level clouds are frequent above the southeast 662 stratocumulus clouds. Atmospheric Chemistry Physics. 20(18). Atlantic and 11025-11043. 663 https://doi.org/10.5194/acp-20-11025-2020 664
  - ARM Data Discovery. (2025). In https://adc.arm.gov/discovery/.
  - Avery, M. A., Ryan, R. A., Getzewich, B. J., Vaughan, M. A., Winker, D. M., Hu, Y., Garnier, A., Pelon, J., & Verhappen, C. A. (2020). CALIOP V4 cloud thermodynamic phase assignment and the impact of near-nadir viewing angles. Atmospheric Measurement Techniques, 13(8), 4539-4563. https://doi.org/10.5194/amt-13-4539-2020
    - Balmes, K. A., Sedlar, J., Riihimaki, L. D., Olson, J. B., Turner, D. D., & Lantz, K. (2023). Regime-Specific Cloud Vertical Overlap Characteristics From Radar and Lidar Observations at the ARM Sites. Journal of Geophysical Research: Atmospheres, 128(6), https://doi.org/10.1029/2022JD037772
    - Barker, H. W., Korolev, A. V., Hudak, D. R., Strapp, J. W., Strawbridge, K. B., & Wolde, M. (2008). A comparison between CloudSat and aircraft data for a multilayer, mixed phase cloud system during the Canadian CloudSat-CALIPSO Validation Project. Journal of Geophysical Research: Atmospheres, 113(D8). https://doi.org/10.1029/2008JD009971
    - Brast, M. and Markmann, P.: Detecting the melting layer with a micro rain radar using a neural network approach, Atmos. Meas. Tech., 13, 6645–6656, https://doi.org/10.5194/amt-13-6645-2020, 2020.
    - Breiman, L. (2001). Random Forests. Machine Learning, 45(1), 5-32. https://doi.org/10.1023/A:1010933404324
    - Cesana, G., & Chepfer, H. (2012). How well do climate models simulate cloud vertical structure? A comparison between CALIPSO-GOCCP satellite observations and CMIP5 models. Geophysical Research Letters, 39(20). https://doi.org/10.1029/2012GL053153
    - Cesana, G., & Chepfer, H. (2013). Evaluation of the cloud thermodynamic phase in a climate model using CALIPSO-GOCCP. Journal of Geophysical Research: Atmospheres, 118(14), 7922-7937. https://doi.org/10.1002/jgrd.50376
    - Cesana, G. V., Ackerman, A. S., Fridlind, A. M., Silber, I., Del Genio, A. D., Zelinka, M. D., Chepfer, H., Khadir, T., & Roehrig, R. (2024). Observational constraint on a feedback from supercooled clouds reduces projected warming uncertainty. Communications Earth & Environment, 5(1), 181-181. https://doi.org/10.1038/s43247-024-01339-1
    - Chollet, F., et al. (2015) Keras. https://keras.io
    - Clothiaux, E., Miller, M., Perez, R., Turner, D., Moran, K., Martner, B., Ackerman, T., Mace, G., Marchand, R., Widener, K., Rodriguez, D., Uttal, T., Mather, J., Flynn, C., Gaustad, K., & Ermold, B. (2001). The ARM Millimeter Wave Cloud Radars (MMCRs) and the Active Remote Sensing of Clouds (ARSCL) Value Added Product (VAP).
    - Curry, J. A., Schramm, J. L., Rossow, W. B., & Randall, D. (1996), Overview of Arctic Cloud and Radiation Characteristics. Journal of Climate, 9(8), 1731-1764. https://doi.org/10.1175/1520-0442(1996)009<1731:OOACAR>2.0.CO;2
    - Fairless, T., Jensen, M., Zhou, A., & Giangrande, S. (2021). Interpolated Sounding and Gridded Sounding Value-Added Products.
    - Fan, J., Ghan, S., Ovchinnikov, M., Liu, X., Rasch, P. J., & Korolev, A. (2011). Representation of Arctic mixed-phase clouds and the Wegener-Bergeron-Findeisen process in climate models: Perspectives from a cloud-resolving study. Journal of Geophysical Research, 116, D00T07-D00T07. https://doi.org/10.1029/2010JD015375
    - Flynn, D., Cromwell, E., & Zhang, D. (2023). Micropulse Lidar Cloud Mask Machine-Learning Value-Added Product Report. Galea, D., Ma, H.-Y., Wu, W.-Y., & Kobayashi, D. (2024). Deep Learning Image Segmentation for Atmospheric Rivers. Artificial Intelligence for the Earth Systems, 3(1). https://doi.org/10.1175/AIES-D-23-0048.1
    - Gaustad, K. L., Turner, D. D., & McFarlane, S. A. (2011). MWRRET Value-Added Product: The Retrieval of Liquid Water Path and Precipitable Water Vapor from Microwave Radiometer (MWR) Data Sets (Revision 2).
    - Geerts, B., Giangrande, S. E., McFarquhar, G. M., Xue, L., Abel, S. J., Comstock, J. M., Crewell, S., DeMott, P. J., Ebell, K., Field, P., Hill, T. C. J., Hunzinger, A., Jensen, M. P., Johnson, K. L., Juliano, T. W., Kollias, P., Kosovic, B., Lackner, C., Luke, E.,... Wu, P. (2022). The COMBLE Campaign: A Study of Marine Boundary Layer Clouds in Arctic Cold-Outbreaks. Bulletin of the American Meteorological Society, 103(5), E1371-E1389. https://doi.org/10.1175/BAMS-D-21-0044.1
- Geiss, A., & Hardin, J. C. (2021). Inpainting radar missing data regions with deep learning. Atmospheric Measurement 707 Techniques, 14(12), https://doi.org/10.5194/amt-14-7729-2021

- Heaton, J. (2018). Ian Goodfellow, Yoshua Bengio, and Aaron Courville: Deep learning: The MIT Press, 2016, 800 pp, ISBN: 0262035618 Article. *Genetic Programming and Evolvable Machines*, 19(1-2).
- Hogan, R. J., Illingworth, A. J., O'Connor, E. J., & Baptista, J. P. V. P. (2003). Characteristics of mixed-phase clouds. II: A climatology from ground-based lidar. *Quarterly Journal of the Royal Meteorological Society*, 129(592), 2117-2134. https://doi.org/10.1256/qj.01.209

- Huang, H., Lin, L., Tong, R., Hu, H., Zhang, Q., Iwamoto, Y., Han, X., Chen, Y. W., & Wu, J. (2020). UNet 3+: A Full-Scale Connected UNet for Medical Image Segmentation. ICASSP, IEEE International Conference on Acoustics, Speech and Signal Processing Proceedings,
- Ioffe, S., & Szegedy, C. (2015). Batch normalization: Accelerating deep network training by reducing internal covariate shift. 32nd International Conference on Machine Learning, ICML 2015,
- Kalesse, H., de Boer, G., Solomon, A., Oue, M., Ahlgrimm, M., Zhang, D., Shupe, M. D., Luke, E., & Protat, A. (2016). Understanding Rapid Changes in Phase Partitioning between Cloud Liquid and Ice in Stratiform Mixed-Phase Clouds: An Arctic Case Study. *Monthly Weather Review*, 144(12), 4805-4826. https://doi.org/10.1175/MWR-D-16-0155.1
- Kalogeras, P., Battaglia, A., and Kollias, P.: Supercooled Liquid Water Detection Capabilities from Ka-Band Doppler Profiling Radars: Moment-Based Algorithm Formulation and Assessment, Remote Sens., 13, 2891, https://doi.org/10.3390/rs13152891, 2021.
- Kay, J. E., & L'Ecuyer, T. (2013). Observational constraints on Arctic Ocean clouds and radiative fluxes during the early 21st century. *Journal of Geophysical Research: Atmospheres*, 118(13), 7219-7236. <a href="https://doi.org/10.1002/jgrd.50489">https://doi.org/10.1002/jgrd.50489</a>
- Kay, J. E., L'Ecuyer, T., Gettelman, A., Stephens, G., & O'Dell, C. (2008). The contribution of cloud and radiation anomalies to the 2007 Arctic sea ice extent minimum. *Geophysical Research Letters*, 35(8). https://doi.org/10.1029/2008GL033451
- King, F., C. Pettersen, C. G. Fletcher, and A. Geiss, 2024: Development of a Full-Scale Connected U-Net for Reflectivity Inpainting in Spaceborne Radar Blind Zones. Artif. Intell. Earth Syst., 3, e230063, https://doi.org/10.1175/AIES-D-23-0063.1.
- Korolev, A., & Milbrandt, J. (2022). How Are Mixed-Phase Clouds Mixed? *Geophysical Research Letters*, 49(18). https://doi.org/10.1029/2022GL099578
- Krizhevsky, A., Sutskever, I., & Hinton, G. E. (2017). ImageNet classification with deep convolutional neural networks. *Communications of the ACM*, 60(6). https://doi.org/10.1145/3065386
- Lackner, C. P., Geerts, B., Juliano, T. W., Kosovic, B., & Xue, L. (2024). Characterizing Mesoscale Cellular Convection in Marine Cold Air Outbreaks With a Machine Learning Approach. *Journal of Geophysical Research: Atmospheres*, 129(14). https://doi.org/10.1029/2024JD041651
- Lagerquist, R., Turner, D. D., Ebert-Uphoff, I., & Stewart, J. Q. (2023). Estimating Full Longwave and Shortwave Radiative Transfer with Neural Networks of Varying Complexity. *Journal of Atmospheric and Oceanic Technology*, 40(11), 1407-1432. https://doi.org/10.1175/JTECH-D-23-0012.1
- LeCun, Y., Bottou, L., Bengio, Y., & Haffner, P. (1998). Gradient-based learning applied to document recognition. *Proceedings of the IEEE*, 86(11). https://doi.org/10.1109/5.726791
- Lenhardt, J., Quaas, J., Sejdinovic, D., and Klocke, D.: CloudViT: classifying cloud types in global satellite data and in kilometre-resolution simulations using vision transformers, EGUsphere [preprint], https://doi.org/10.5194/egusphere-2024-2724, 2024.
- Lu, Y., & Kumar, J. (2019, 2019/11//). Convolutional Neural Networks for Hydrometeor Classification using Dual Polarization Doppler Radars. 2019 International Conference on Data Mining Workshops (ICDMW),
- Luke, E. P., Kollias, P., and Shupe, M. D.: Detection of supercooled liquid in mixed-phase clouds using radar Doppler spectra, J. Geophys. Res. Atmos., 115, D19201, https://doi.org/10.1029/2009JD012884, 2010.
- McFarquhar, G. M., Ghan, S., Verlinde, J., Korolev, A., Strapp, J. W., Schmid, B., Tomlinson, J. M., Wolde, M., Brooks, S. D., Cziczo, D., Dubey, M. K., Fan, J., Flynn, C., Gultepe, I., Hubbe, J., Gilles, M. K., Laskin, A., Lawson, P., Leaitch, W. R.,...Glen, A. (2011). Indirect and Semi-direct Aerosol Campaign. *Bulletin of the American Meteorological Society*, 92(2), 183-201. https://doi.org/10.1175/2010BAMS2935.1
- Mülmenstädt, J., O. Sourdeval, J. Delanoë, J., and Quaas, J.: Frequency of Occurrence of Rain from Liquid-, Mixed-, and Ice-Phase Clouds Derived from A-Train Satellite Retrievals, Geophys. Res. Lett., 42, 6502–6509, https://doi.org/10.1002/2015GL064604, 2015.

Pedregosa, F., Varoquaux, G., Gramfort, A., Michel, V., Thirion, B., Grisel, O., Blondel, M., Prettenhofer, P., Weiss, R., Dubourg, V., Vanderplas, J., Passos, A., Cournapeau, D., Brucher, M., Perrot, M., & Duchesnay, É. (2011). Scikit-learn: Machine learning in Python. *Journal of Machine Learning Research*, 12.

- Pithan, F., Medeiros, B., and Mauritsen, T.: Mixed-phase clouds cause climate model biases in Arctic wintertime temperature inversions, Clim. Dynam., 43, 289–303, https://doi.org/10.1007/s00382-013-1964-9, 2014.
  - Platnick, S., Meyer, K. G., King, M. D., Wind, G., Amarasinghe, N., Marchant, B., Arnold, G. T., Zhang, Z., Hubanks, P. A., Holz, R. E., Yang, P., Ridgway, W. L., and Riedi, J.: The MODIS Cloud Optical and Microphysical Products: Collection 6 Updates and Examples From Terra and Aqua, IEEE T. Geosci. Remote, 55, 502–525, <a href="https://doi.org/10.1109/TGRS.2016.2610522">https://doi.org/10.1109/TGRS.2016.2610522</a>, 2017.
  - Ronneberger, O., Fischer, P., & Brox, T. (2015). U-net: Convolutional networks for biomedical image segmentation. Lecture Notes in Computer Science (including subseries Lecture Notes in Artificial Intelligence and Lecture Notes in Bioinformatics),
  - Schimmel, W., Kalesse-Los, H., Maahn, M., Vogl, T., Foth, A., Garfias, P. S., and Seifert, P.: Identifying cloud droplets beyond lidar attenuation from vertically pointing cloud radar observations using artificial neural networks, Atmos. Meas. Tech., 15, 5343–5366, https://doi.org/10.5194/amt-15-5343-2022, 2022.
  - Sha, Y., Gagne, D. J., West, G., & Stull, R. (2020). Deep-learning-based gridded downscaling of surface meteorological variables in complex terrain. Part i: Daily maximum and minimum 2-m temperature. *Journal of Applied Meteorology and Climatology*, 59(12). https://doi.org/10.1175/JAMC-D-20-0057.1
  - Shupe, M. D. (2007). A ground-based multisensor cloud phase classifier. *Geophysical Research Letters*, 34(22). https://doi.org/10.1029/2007GL031008
  - Shupe, M. D., Walden, V. P., Eloranta, E., Uttal, T., Campbell, J. R., Starkweather, S. M., and Shiobara, M.: Clouds at Arctic Atmospheric Observatories. Part I: Occurrence and Macrophysical Properties, J. Appl. Meteorol. Clim., 50, 626–644, https://doi.org/10.1175/2010JAMC2467.1, 2011.
  - Shupe, M. D. (2011). Clouds at Arctic Atmospheric Observatories. Part II: Thermodynamic Phase Characteristics. *Journal of Applied Meteorology and Climatology*, 50(3), 645-661. https://doi.org/10.1175/2010JAMC2468.1
  - Shupe, M. D., Comstock, J. M., Turner, D. D., & Mace, G. G. (2016). Cloud Property Retrievals in the ARM Program. Meteorological Monographs, 57, 19.11-19.20. https://doi.org/10.1175/AMSMONOGRAPHS-D-15-0030.1
  - Shupe, M. D., & Intrieri, J. M. (2004). Cloud Radiative Forcing of the Arctic Surface: The Influence of Cloud Properties, Surface Albedo, and Solar Zenith Angle. *Journal of Climate*, 17(3), 616-628. <a href="https://doi.org/10.1175/1520-0442(2004)017<0616:CRFOTA>2.0.CO;2">https://doi.org/10.1175/1520-0442(2004)017<0616:CRFOTA>2.0.CO;2</a>
  - Shupe, M. D., Turner, D. D., Zwink, A., Thieman, M. M., Mlawer, E. J., & Shippert, T. (2015). Deriving Arctic Cloud Microphysics at Barrow, Alaska: Algorithms, Results, and Radiative Closure. *Journal of Applied Meteorology and Climatology*, 54(7), 1675-1689. https://doi.org/10.1175/JAMC-D-15-0054.1
  - Silber, I., Verlinde, J., Eloranta, E. W., & Cadeddu, M. (2018). Antarctic cloud macrophysical, thermodynamic phase, and atmospheric inversion coupling properties at McMurdo Station: I. Principal data processing and climatology. *Journal of Geophysical Research: Atmospheres*, 123, 6099–6121. https://doi.org/10.1029/2018JD028279.
  - Silber, I., Verlinde, J., Wen, G., & Eloranta, E. W. (2020). Can Embedded Liquid Cloud Layer Volumes Be Classified in Polar Clouds Using a Single- Frequency Zenith-Pointing Radar? *IEEE Geoscience and Remote Sensing Letters*, 17(2), 222-226. https://doi.org/10.1109/LGRS.2019.2918727
  - Silber, I., Fridlind, A. M., Verlinde, J., Ackerman, A. S., Cesana, G. V., & Knopf, D. A. (2021). The prevalence of precipitation from polar supercooled clouds. *Atmospheric Chemistry and Physics*, 21(5), 3949-3971. <a href="https://doi.org/10.5194/acp-21-3949-2021">https://doi.org/10.5194/acp-21-3949-2021</a>
  - Solomon, A., de Boer, G., Creamean, J. M., McComiskey, A., Shupe, M. D., Maahn, M., & Cox, C. (2018). The relative impact of cloud condensation nuclei and ice nucleating particle concentrations on phase partitioning in Arctic mixed-phase stratocumulus clouds. *Atmospheric Chemistry and Physics*, 18(23), 17047-17059. <a href="https://doi.org/10.5194/acp-18-17047-2018">https://doi.org/10.5194/acp-18-17047-2018</a>
  - Solomon, A., Shupe, M. D., Persson, P. O. G., & Morrison, H. (2011). Moisture and dynamical interactions maintaining decoupled Arctic mixed-phase stratocumulus in the presence of a humidity inversion. *Atmospheric Chemistry and Physics*, 11(19), 10127-10148. <a href="https://doi.org/10.5194/acp-11-10127-2011">https://doi.org/10.5194/acp-11-10127-2011</a>

- Springenberg, M., Frommholz, A., Wenzel, M., Weicken, E., Ma, J., & Strodthoff, N. (2023). From modern CNNs to vision transformers: Assessing the performance, robustness, and classification strategies of deep learning models in histopathology. *Medical Image Analysis*, 87. <a href="https://doi.org/10.1016/j.media.2023.102809">https://doi.org/10.1016/j.media.2023.102809</a>
- Srivastava, N., Hinton, G., Krizhevsky, A., Sutskever, I., & Salakhutdinov, R. (2014). Dropout: A simple way to prevent neural networks from overfitting. *Journal of Machine Learning Research*, 15.
- Storelymo, T. and Tan, I.: The Wegener-Bergeron-Findeisen Process Its Discovery and Vital Importance for Weather and Climate, Meteorol. Z., 24, 455–461, https://doi.org/10.1127/metz/2015/0626, 2015.

- Taghanaki, S., Zheng, Y., Zhou, K., Georgescu, B., Sharma, P., Xu, D., Comaniciu, D., Hamarneh, G. (2019). Combo Loss: Handling Input and Output Imbalance in Multi-Organ Segmentation. *Science Direct*, 75(24-33). https://doi.org/10.1016/j.compmedimag.2019.04.005
- Tan, I., Storelvmo, T., & Zelinka, M. D. (2016). Observational constraints on mixed-phase clouds imply higher climate sensitivity. *Science*, 352(6282), 224-227. https://doi.org/10.1126/science.aad5300.
- Tan, I., Zhou, C., Lamy, A., and Stauffer, C. L.: Moderate climate sensitivity due to opposing mixed-phase cloud feedbacks, Clim. Atmos. Sci., 8, 86, doi:10.1038/s41612-025-00948-7, 2025.
- Thisanke, H., Deshan, C., Chamith, K., Seneviratne, S., Vidanaarachchi, R., & Herath, D. (2023). Semantic segmentation using Vision Transformers: A survey. Engineering Applications of Artificial Intelligence, 126, 106669. https://doi.org/10.1016/j.engappai.2023.106669.
- Turner, D. D., Ackerman, S. A., Baum, B. A., Revercomb, H. E., & Yang, P. (2003). Cloud Phase Determination Using Ground-Based AERI Observations at SHEBA. *Journal of Applied Meteorology*, 42(6), 701-715. https://doi.org/10.1175/1520-0450(2003)042<0701:CPDUGA>2.0.CO;2
- Van Weverberg, K., Giangrande, S., Zhang, D., Morcrette, C. J., & Field, P. R. (2023). On the Role of Macrophysics and Microphysics in Km-Scale Simulations of Mixed-Phase Clouds During Cold Air Outbreaks. *Journal of Geophysical Research: Atmospheres*, 128(11). https://doi.org/10.1029/2022JD037854
- Veillette, M. S., Kurdzo, J. M., Stepanian, P. M., McDonald, J., Samsi, S., & Cho, J. Y. N. (2023). A Deep Learning–Based Velocity Dealiasing Algorithm Derived from the WSR-88D Open Radar Product Generator. *Artificial Intelligence for the Earth Systems*, 2(3). https://doi.org/10.1175/AIES-D-22-0084.1
- Verlinde, J., Harrington, J. Y., McFarquhar, G. M., Yannuzzi, V. T., Avramov, A., Greenberg, S., Johnson, N., Zhang, G., Poellot, M. R., Mather, J. H., Turner, D. D., Eloranta, E. W., Zak, B. D., Prenni, A. J., Daniel, J. S., Kok, G. L., Tobin, D. C., Holz, R., Sassen, K.,...Schofield, R. (2007). The Mixed-Phase Arctic Cloud Experiment. Bulletin of the American Meteorological Society, 88(2), 205-222. https://doi.org/10.1175/BAMS-88-2-205
- Verlinde, J., Zak, B. D., Shupe, M. D., Ivey, M. D., & Stamnes, K. (2016). The ARM North Slope of Alaska (NSA) Sites. *Meteorological Monographs*, *57*, 8.1-8.13. <a href="https://doi.org/10.1175/AMSMONOGRAPHS-D-15-0023.1">https://doi.org/10.1175/AMSMONOGRAPHS-D-15-0023.1</a>
- Wang, Z., & Sassen, K. (2001). Cloud Type and Macrophysical Property Retrieval Using Multiple Remote Sensors. *Journal of Applied Meteorology*, 40(10), 1665-1682. https://doi.org/10.1175/1520-0450(2001)040<1665:CTAMPR>2.0.CO:2
- Wendisch, M., Macke, A., Ehrlich, A., Lüpkes, C., Mech, M., Chechin, D., Dethloff, K., Velasco, C. B., Bozem, H., Brückner, M., Clemen, H.-C., Crewell, S., Donth, T., Dupuy, R., Ebell, K., Egerer, U., Engelmann, R., Engler, C., Eppers, O.,...Zeppenfeld, S. (2019). The Arctic Cloud Puzzle: Using ACLOUD/PASCAL Multiplatform Observations to Unravel the Role of Clouds and Aerosol Particles in Arctic Amplification. Bulletin of the American Meteorological Society, 100(5), 841-871. https://doi.org/10.1175/BAMS-D-18-0072.1
- Weyn, J. A., Durran, D. R., Caruana, R., & Cresswell-Clay, N. (2021). Sub-Seasonal Forecasting With a Large Ensemble of Deep-Learning Weather Prediction Models. *Journal of Advances in Modeling Earth Systems*, 13(7). <a href="https://doi.org/10.1029/2021MS002502">https://doi.org/10.1029/2021MS002502</a>
- Wieland, M., Li, Y., & Martinis, S. (2019). Multi-sensor cloud and cloud shadow segmentation with a convolutional neural network. *Remote Sensing of Environment*, 230. https://doi.org/10.1016/j.rse.2019.05.022
- Xia, Z., & McFarquhar, G. M. (2024). Dependence of Cloud Macrophysical Properties and Phase Distributions on Environmental Conditions Over the North Atlantic and Southern Ocean: Results From COMBLE and MARCUS. Journal of Geophysical Research: Atmospheres, 129(12). https://doi.org/10.1029/2023JD039869
  - Xie, Y., King, F., Pettersen, C., & Flanner, M. (2025). Machine learning detection of melting layers from radar observations. *Journal of Geophysical Research: Machine Learning and Computation*, 2(2), e2024JH000521.

Yu, G., Verlinde, J., Clothiaux, E. E., and Chen, Y.-S.: Mixed-phase cloud phase partitioning using millimeter wavelength cloud radar Doppler velocity spectra, J. Geophys. Res.-Atmos., 119, 7556–7576, https://doi.org/10.1002/2013JD021182, 2014.

- Zhang, D., & Levin, M. (2024). Thermodynamic cloud phase (THERMOCLDPHASE), 2017-03-01 to 2024-07-01, North Slope Alaska (NSA), Central Facility, Barrow AK (C1). In *Atmospheric Radiation Measurement (ARM) User Facility*.
  - Zhang, D., Wang, Z., & Liu, D. (2010). A global view of midlevel liquid-layer topped stratiform cloud distribution and phase partition from CALIPSO and CloudSat measurements. *Journal of Geophysical Research: Atmospheres*, 115(D4). https://doi.org/10.1029/2009JD012143
  - Zhang, D., Wang, Z., Luo, T., Yin, Y., & Flynn, C. (2017). The occurrence of ice production in slightly supercooled Arctic stratiform clouds as observed by ground-based remote sensors at the ARM NSA site. *Journal of Geophysical Research: Atmospheres*, 122(5), 2867-2877. https://doi.org/10.1002/2016JD026226
  - Zheng, X., Tao, C., Zhang, C., Xie, S., Zhang, Y., Xi, B., & Dong, X. (2023). Assessment of CMIP5 and CMIP6 AMIP Simulated Clouds and Surface Shortwave Radiation Using ARM Observations over Different Climate Regions. *Journal of Climate*, 36(24), 8475-8495. https://doi.org/10.1175/JCLI-D-23-0247.1
  - Zhou, Z., Rahman Siddiquee, M. M., Tajbakhsh, N., & Liang, J. (2018). Unet++: A nested u-net architecture for medical image segmentation. Lecture Notes in Computer Science (including subseries Lecture Notes in Artificial Intelligence and Lecture Notes in Bioinformatics),