# Peer review of "Classifying Thermodynamic Cloud Phase Using Machine Learning"

_EGUsphere, 2025_

## Author Comment (AC1)

**Reply to Reviewer #1's comments**

*This work addresses an important aspect of polar clouds based on the recent THERMODCLDPHASE VAP, elucidates aspects of its algorithm, and creatively investigates how to extend it using machine learning. I believe their product will be a useful contribution to the scientific community. My only main concern is the relatively minor role that lidar plays in the phase classifications and therefore the relatively poor skill that the models have in predicting liquid. I suggest that the authors delve further into how liquid phase predictions can be improved while removing the somewhat redundant parts of the manuscript describing the drop-out experiments described below. The results also seem to show that temperature plays a more dominant role than shown by the analysis methods. I would recommend publication of this manuscript after the authors consider the suggestions below.*

**Answer: We thank the reviewer for these suggestions and comments. We carefully revised the manuscript according to the reviewer's comments.**

*Clarifying the geographical scope of the work*

- *It is unclear whether the scope of the THERMOCLDPHASE VAP is limited to Arctic (or polar) clouds. Different regions may require different tuning in the algorithm, and the authors have only focused on the Arctic (NSA and COMBLE regions) in this manuscript. If the goal is to test the generalization of the machine learning models to other regions, does this also include different cloud types? Is THERMOCLDPHASE suited for classifying the thermodynamic phase of other types of clouds? Line 57 mentions "several other ARM observatories across the world" but does not specify which. I recommend that the authors clarify this in the manuscript.*

**Answer: We thank the reviewer for this valuable suggestion. We agree that algorithm tuning may need to be region-specific. As noted by Shupe (2006), the multi-sensor cloud thermodynamic phase classification method "has been specifically developed for observations of Arctic clouds." Accordingly, the THERMOCLDPHASE VAP is currently applied only at the seven ARM high-latitude observatories. We have revised the sentence in line 60 to read: "as well as six other ARM high-latitude observatories." Additionally, since the algorithm does not include the classification of hail and graupel, it has difficulties distinguishing these hydrometeor types in deep convective cloud regimes over tropical and mid-latitude regions. To improve clarity, we added the following sentence between lines 60 and 63: "It is noted that the multi-sensor cloud thermodynamic phase classification was specifically developed for observations of Arctic clouds (Shupe 2006). Since the algorithm does not include the classification of hail and graupel, it has difficulty distinguishing these hydrometeor types in deep convective cloud regimes over tropical and mid-latitude regions."**

*Clarifying the roles of temperature and lidar*

- *The sharp cut-off along the 0˚C isotherm in Figure 1(g) where ice transitions to warm precipitation suggests to me that temperature plays a critical role in determining the phase of hydrometeors. Yet, it seems that the "feature importance analysis" and Figure 8 show that radar plays an even more important role than temperature. How do the authors reconcile this?*

**Answer: It is true that temperature plays a critical role in determining hydrometeor phase within the transition region from ice to warm precipitation. However, this transition typically occurs within a relatively narrow vertical layer. Outside of this zone—particularly at temperatures above 0 °C or below –40 °C—temperature becomes less influential, and radar measurements provide more definitive information for hydrometeor phase classification.**

- *Also, I am concerned about the minor role that lidar plays in the phase classification presumably due to lidar attenuation. The fact that the CNN performs the best out of the three models due to its accurate prediction of ice makes sense in light of the fact that radar and temperature play the most important roles in the prediction --- radar can better observe the ice particles that are larger in size, and ice crystals that freeze homogeneously are easier to identify. I suggest that the authors separately show cases that are dominated by single-layer thin liquid clouds to check whether lidar plays a more significant role and whether the models might also show high fidelity for the liquid classifications.*

**Answer: We appreciate the reviewer's concern and thoughtful suggestions. As noted in previous studies (Shupe et al., 2011; Zhang et al., 2017), lidar observations at the NSA site are often fully attenuated above approximately 1 km due to persistent low-level clouds. This limitation contributes to the generally low feature importance of lidar data in thermodynamic phase classification across the full dataset.**

**However, for liquid-phase identification specifically, lidar backscatter shows notable importance in the CNN model. While lidar backscatter and depolarization ratio offer direct and reliable indicators of liquid-phase presence, radar-based variables—such as reflectivity, mean Doppler velocity, and spectral width—can also contain useful signatures of liquid-phase clouds (Luke et al., 2010; Yu et al., 2014; Kalogeras et al., 2021; Schimmel et al., 2022). This is consistent with our feature importance results presented in Figure 8.**

**We have added several sentences in line between 439 and 442 to further clarify this point. As suggested, a dedicated analysis of single-layer thin liquid clouds could help better assess the role of lidar in these specific conditions. Developing a refined training dataset focused on such cloud regimes and retraining the models would be a valuable extension of this work, and we consider this a promising direction for future research.**

**References:**

Kalogeras, P., Battaglia, A., and Kollias, P.: Supercooled Liquid Water Detection Capabilities from Ka-Band Doppler Profiling Radars: Moment-Based Algorithm Formulation and Assessment, Remote Sens., 13, 2891, https://doi.org/10.3390/rs13152891, 2021.

Luke, E. P., Kollias, P., and Shupe, M. D.: Detection of supercooled liquid in mixed-phase clouds using radar Doppler spectra, J. Geophys. Res. Atmos., 115, D19201, https://doi.org/10.1029/2009JD012884, 2010.

Schimmel, W., Kalesse-Los, H., Maahn, M., Vogl, T., Foth, A., Garfias, P. S., and Seifert, P.: Identifying cloud droplets beyond lidar attenuation from vertically pointing cloud radar observations using artificial neural networks, Atmos. Meas. Tech., 15, 5343–5366, https://doi.org/10.5194/amt-15-5343-2022, 2022.

Shupe, M. D., Walden, V. P., Eloranta, E., Uttal, T., Campbell, J. R., Starkweather, S. M., and Shiobara, M.: Clouds at Arctic Atmospheric Observatories. Part I: Occurrence and Macrophysical Properties, J. Appl. Meteorol. Clim., 50, 626–644, https://doi.org/10.1175/2010JAMC2467.1, 2011.

Yu, G., Verlinde, J., Clothiaux, E. E., and Chen, Y.-S.: Mixed-phase cloud phase partitioning using millimeter wavelength cloud radar Doppler velocity spectra, J. Geophys. Res.-Atmos., 119, 7556–7576, https://doi.org/10.1002/2013JD021182, 2014.

*Suggestions with regards to writing:*

*Abstract:*

- *The Abstract does not mention what the results for COMBLE are.*

**Answer: We added a sentence in line 26 to read: "The models demonstrated similar performance to that observed at the NSA site."**

- *Similarly, the results of the ML models' response to simulated instrument outages and signal degradation are also not summarized in the Abstract.*

**Answer: We added a sentence in line 27-28: "and show that CNN U-NET model with input channel dropouts during training performs better when input fields are missing".**

*Introduction:*

- *Lines 31-33 require references. Suggestions: for ice particle production (via the WBF process ---Storlevmo & Tan 2015 ), precipitation formation (Mulmenstadt et al. 2015), the evolution of the cloud life cycle (Pithan et al. 2014), and also the response of clouds to global warming (Tan et al. 2025).*

**Answer:  We are grateful to the reviewer for suggesting these valuable references; they have been included in the revised manuscript.**

- *Lines 36-38: satellite remote sensing could also be included here, e.g. MODIS cloud retrievals as detailed in Platnick et al. (2016).*

**Answer: We agree with the reviewer that satellite-based remote sensing of clouds is important and have added the suggested reference accordingly.**

*Concerns regarding redundancy:*

- *Section 4 essentially shows what the feature importance analysis did earlier in the manuscript regarding the importance of radar and the temperature soundings for the phase predictions. The authors might want to consider removing this section and replacing it with more detailed analysis on the limited role of lidar and how the classification of liquid pixels can be improved.*

**Answer: We appreciate the reviewer's thoughtful suggestion. While we agree that Section 4 reinforces conclusions drawn from the earlier feature importance analysis, it also provides additional value by quantitatively assessing the impact of missing observational inputs on ML model performance. This section offers a more detailed examination of how model predictions degrade in the absence of specific sensors and evaluates the relative resilience of different ML models to missing data. We consider this analysis to be a key contribution of the study, particularly in the context of real-world applications where data gaps are common. Therefore, we have chosen to retain Section 4 in the manuscript.**

*Minor/typographical suggestions:*

- *Please clarify what is meant by "pixel" and "voxel" and also be consistent with the terminology throughout.*

**Answer: We changed "voxel" to "pixel" for consistency throughout the manuscript.**

- *Line 41: "imagers" not "images"?*

**Answer: We retained the term "images" since the context refers to using the captured data for identifying cloud phase. To enhance clarity, we added the word "captured" before "particle images."**

- *Line 104: no dash necessary in "reads-in"*

**Answer: We removed the dash for consistency and clarity.**

- *Please consider including isotherms in panels (a) – (e) as well.*

**Answer: We have updated the figure to include isotherms for these panels.**

- *Line 158: what is the percentage of "unknown" pixels in the VAP?*

**Answer:** Based on one year of data from 2021 at the NSA site, 5.9% of cloud hydrometeors were classified as unknown. We added a sentence at line 168-169 to note this.

- *Table 1: Please define "clip".*

**Answer:** We added a note about the clip function in the caption of Table 1.

- *Lines 237-238: Why was the data limited to only 2018-2020 instead of the full record going back to 1998?*

**Answer:** The ARM THERMOCLDPHASE VAP was processed only for data collected after 2017 at the NSA site, focusing on more recent and reliable lidar and radar measurements (https://adc.arm.gov/discovery/#/results/s::thermocldphase). As discussed in Section 2.3, the existing training dataset is sufficiently large to support ML model development. While including additional data from the NSA site might offer slight improvements in model performance, it would significantly increase the training time.

- *Line 170: "imbalanced" in place of "imbalance"?*

**Answer:** We changed 'imbalance' to imbalanced'.

- *Line 459: CCN should be CNN*

**Answer:** We changed 'CCN' to 'CNN' in the manuscript.

- *Line 252: "In the" in front of "Future"?*

**Answer:** We added 'In the' in front of 'Future' as suggested.

- *Line 284: apostrophe after "models"*

**Answer:** We added an apostrophe after 'models' as suggested.

- *Figure 5: Setting the max of the y-axis to 80 may help enhance the visibility of the drizzle/liq_driz/rain categories.*

**Answer:** We appreciate the reviewer's suggestion and have adjusted the maximum value of the y-axis to 80 accordingly.

---

## Author Comment (AC2)

*Reply to Reviewer #2's comments*

*Goldberger et al. provides a detailed analysis of cloud phase classification using a suite of increasingly sophisticated machine learning (ML) models. The authors find that the traditional threshold-based approach (i.e., the THERMOCLDPHASE VAP) was limited in its ability to classify phase across varying regional climates due to the use of static thresholds, and lacked a general robustness when some inputs were missing (e.g. due to instrument outages). Shifting to an ML-based approach improved classification accuracy at two locations, and provided a framework for more easily understanding variable importance and uncertainty quantification in their predictions. The paper is well structured, and includes many of the minor, yet fundamental, points (e.g.., information about data standardization, splitting, model complexity) that are often missed in these types of manuscripts/projects. I also appreciated the additional work provided in the Supplement detailing the hyperparameter search space and dropout tests. I don't see any critical methodological issues here, the content aligns well with the journal's mission, and I believe this paper would be of great interest to the readers of Atmospheric Measurement Techniques (AMT). However, I do have a few minor points below that I'd like to see addressed on before I fully recommend the paper for publication.*

**Answer: We appreciate the reviewer's valuable suggestions and comments. We have carefully revised the manuscript in accordance with the feedback.**

*General Comments:*

1. *While the performance of the ML models across both sites were mostly consistent, it was clear that the CNN really struggled with liquid/mixed-phase at ANX compared to NSA. Do the authors have more of a physical basis for this drop in accuracy? You discuss this issue briefly in 451-462, but I am curious if the regional cloud structures are different enough in Norway to cause the CNN to struggle for this very important classification category. On a similar note, while performance for other classes remained highest in the CNN, I found the MLP to often provide a consistent performance more generally across all classes (not only in Fig. 9, but also Fig. 6). For instance, in Fig. 6, the CNN really excels at ice-based classifications, but it is slightly worse in nearly every other class compared to the MLP/RF. Some may argue that a less complex model that is more consistent and quicker to train would be preferred compared to the CNN, even with worse performance in some categories.*

**Answer: We thank the reviewer for these insightful comments. We agree that the regional differences in cloud structures likely play a key role in the CNN's reduced performance at ANX, particularly for the liquid and mixed-phase classifications. We attribute this degradation primarily to the more complex and vertically developed convective cloud structures observed during cold-air outbreaks (CAOs) at ANX, which pose greater challenges for classification. A sentence has been added between lines 488 and 489 to clarify this point.**

**As noted in lines 265–278, the CNN's strong performance in ice-phase classification is largely due to imbalanced training data, where ice-phase pixels dominate. We also discussed the challenges of applying balanced training data in CNNs. Additionally, in lines 371–377, we examined how imbalanced training data affect the performance of RF and MLP models.**

2. *General comments about figure quality. As this is a visual project, I feel that the visual comparisons between methods are key for demonstrating performance differences to the reader. For instance, on Fig. 3, could you not zoom in more on the actual cloud structures? The plots extend to 12 km currently, but there is little-to-no activity above say 7 km (and similar for Fig. 9/10/11) which leaves a lot of blank space. Additionally, I like the point being made by Fig. 4, but it is quite challenging to read in its current state. I would make this a 4x2 plot instead of 2x4 and try to make the bars wider. I recognize that the versions of these images I am seeing is also compressed, but to highlight fine structural details, I'd recommend the authors make sure their resolution is as high as is allowed by the journal so readers can zoom in to see the interesting structural details better.*

**Answer: We appreciate the reviewer's suggestions for improving the figure quality. In response, we limited the vertical axis to 8–9 km, just above the top of the highest cloud, in Figures 3, 9, 10, and 11. Additionally, we revised Figures 4 and S5 to a 4×2 layout as recommended. We will upload the highest-resolution versions of all figures permitted by the publication guidelines.**

*Specific Comments:*

- *Lines 18-19: I like the evolution from decision-tree based methods to more advanced NNs, but was curious if you had considered vision transformers for this problem? They've seen quite a surge in popularity over the past few years especially in areas like remote sensing (e.g., Lenhardt et al., 2024 for clouds, or Thisanke et al., 2023 more generally), and while likely beyond the scope of this project, might be of interest down the road as they can pick up on global features often missed by the U-Net.*

**Answer: We thank the reviewer for highlighting these recent studies that apply advanced neural networks. As noted in lines 648–650, we mentioned that more advanced segmentation algorithms, including vision transformers, could be explored in future work. We have now added the suggested recent studies to the manuscript as recommended.**

- *Lines 177: I appreciate the authors providing a list of RF hyperparameters, and I was curious about using 20 as the maximum depth here. 20 is a fairly deep tree for 100k samples and I am wondering how sensitive the model is to changing this value to something smaller (10-15).*

**Answer: Our initial experiments with the RF model showed that its performance did not change substantially with hyperparameter adjustments, and the best validation performance was achieved using the default scikit-learn hyperparameters. We have added a sentence in lines 190–192 to note this.**

- *Line 185: Interesting, I've never seen someone use the scikit implementation to train an MLP (usually torch or tensorflow)! I'm guessing you used one of these for the CNN?*

**Answer: We appreciate the reviewer's observation. We have updated the text to clarify that TensorFlow, through the Keras interface, was used for the CNN implementation.**

- *Lines 202-203: How are missing pixels handled before being ingested by the CNN? Like what if you have to QC a small patch of bad radar data and mark it as missing, is that whole scene dropped?*

**Answer: Missing instrument data values are filled with a value of –9999 prior to dataset normalization, as specified in Table 1, which results in clipping to the bottom of their valid input range. On the other hand, if the ground truth labels for any batch of data are missing or classified as unknown, the entire batch is discarded and not used to train the model. We added a sentence in line 217-220 to clarify this.**

- *Figure 3: Zooming in on e-g of Fig. 3, we can see the melting layer is one of the most challenging areas to accurately predict for the CNN, with low confidence throughout. As this is an area where change is quickly happening to particle phase, shape and fallspeed, the CNN appears to struggle in resolving these fine scale details that are quite important (the presence, location, and width of the ML are often some of the most important features we would like to accurately classify for looking at regime shifts). Do you have any insights on how to improve this detection here? There have been some projects using ML to better predict this region (e.g., Brast and Markmann, 2020), but it remains an active area of research.*

**Answer: We appreciate the reviewer's thoughtful observation regarding the CNN's difficulty in accurately classifying the melting layer. We agree that this region—where rapid transitions in particle phase, shape, and fall speed occur—is particularly important and challenging to resolve. In our case, the limitations of the THERMOCLDPHASE VAP, including its coarse vertical resolution and reliance on interpolated sounding data, contribute to the reduced model confidence in this layer. As a result, all ML models show diminished performance in this region. We concur that improving detection in the melting layer will likely require a refined training dataset that specifically targets this transitional zone. As the reviewer noted, this remains an active area of research. We have added a sentence between lines 303-307 to reflect this discussion.**

- *Figure 8: While I think this is a useful figure, it is a bit busy with how it is currently set up and takes a moment to digest (variable by phase by model). Do you have an overall feature importance chart anywhere to give a general overview of each input's importance across all phase types?*

**Answer: We appreciate the reviewer's helpful observations regarding Figure 8. We agree that the figure is somewhat busy and may take a moment to interpret. We have updated the figure to improve visual separation of the features and to make the feature labels clearer.**

Because of the cold temperatures at NSA, warm cloud phase categories represent extreme minority classes. As a result, computing overall feature importance based on the change in multi-class recall from input permutations would largely reflect the importance for the dominant ice category. Given this class imbalance, it is important to analyze feature importance separately for each phase. For this reason, we believe it remains valuable to present phase-specific results in Figure 8. For the reviewer's reference, we have also provided a figure below showing the average permutation importance for each feature in the CNN model, calculated as the mean across phase-specific importances.

[Figure]

- *Table 4: I see in Table 4 that you experiment with leaving out different variables, but did you perform retraining on these models without the combination of MPL/LWP variables which display little relevance in Fig. 8? It would appear as though they don't provide much useful context beyond what is given in the other variables.*

Answer: As noted in lines 424–426, "input features from lidar measurements (panels a, e, and i) and the MWRRET LWP (panels c, g, and k) are less influential, likely because lidar signals are often attenuated by persistent low-level clouds at the NSA site (Shupe et al., 2011; Zhang et al., 2017), and LWP provides only column-integrated information rather than detailed vertical profiles." While lidar backscatter and depolarization ratio provide direct and reliable indicators of liquid-phase presence, radar-based variables—such as reflectivity, mean Doppler velocity, and spectral width—also contain useful signatures of liquid-phase clouds. However, in situations where radar data are missing, MPL and LWP can become critical for

cloud phase classification. Moreover, as shown in Figure 8, including MPL and LWP does contribute to improving model predictions in certain scenarios. Therefore, we did not excluded these variables.

- *Section 3.3: While not critical to the current manuscript, did you consider looking at feature/saliency maps in the CNN (e.g., Haar et al., 2023)? As a vision model, this can sometimes be useful for sanity checks that the model is looking at regions in the data that we would expect from a physical standpoint as being relevant (e.g. cloud edges, hydrometeor gradients, gaps etc.).*

Answer: We appreciate the reviewer's valuable suggestion. We agree that incorporating feature or saliency map analysis is a promising avenue for future work to enhance the interpretability of model predictions. We have added a sentence and the suggested reference in lines 648–650, which reads: "*Furthermore, feature or saliency map analysis could offer valuable insights into whether the CNN focuses on physically meaningful regions of the data, and represents a promising direction for future work to enhance the interpretability of model predictions (Haar et al., 2023)."* In addition to verifying that the model is looking at appropriate regions in the input data, this type of analysis could also help determine how much spatial information the model is using and how large the perceptive field needs to be, which could aid in future model development.

*Section 3.4: It is great to see the model also tested at an independent site. I touched on this in my general comment above, but how well do you believe this model could generalize to other regional climates? For instance, other DOE ARM sites like SBS/SGP/ENA or even AWR with similar instrument setups?*

Answer: As noted by Shupe (2007), the multi-sensor cloud thermodynamic phase classification method "has been specifically developed for observations of Arctic clouds." Accordingly, the THERMOCLDPHASE VAP is currently applied only at the seven ARM high-latitude observatories including AWR. We have revised the sentence on line 57 to read: "as well as six other ARM high-latitude observatories." Additionally, since the algorithm does not include the classification of hail and graupel, it has difficulties distinguishing these hydrometeor types in deep convective cloud regimes over tropical and mid-latitude regions. To improve clarity, we added the following sentence between lines 57 and 60: "It is noted that the multi-sensor cloud thermodynamic phase classification was specifically developed for observations of Arctic clouds (Shupe 2007). Since the algorithm does not include the classification of hail and graupel, it has difficulty distinguishing these hydrometeor types in deep convective cloud regimes over tropical and mid-latitude regions.".

- *Line 459: This should be CNN not CCN I believe.*

Answer: We changed 'CCN' to 'CNN' in the manuscript.

- *Line 602: Advanced U-Net models like these have already started to be used in recent literature at other DOE sites looking at clouds using radar data (e.g., King et al., 2025).*

**Answer: We appreciate the reviewer for pointing out the recent literature. We added the reference in line 647.**

- *Line 612: The GitHub repo link appears to be broken here. It would be great to also have a Google Colab notebook available to reproduce some of the model results if possible.*

**Answer: We appreciate the reviewer's suggestions for reproducibility of our results. We have fixed the link and made the GitHub repo public. The data are too large for Google Colab, but we have included code to download the data from ARM and all of the code used to generate results and figures for the manuscript.**

References

Brast, M. and Markmann, P.: Detecting the melting layer with a micro rain radar using a neural network approach, Atmos. Meas. Tech., 13, 6645–6656, https://doi.org/10.5194/amt-13-6645-2020, 2020.

Haar, L. V., Elvira, T., & Ochoa, O. (2023). An analysis of explainability methods for convolutional neural networks. Engineering Applications of Artificial Intelligence, 117, 105606. https://doi.org/10.1016/j.engappai.2022.105606

King, F., C. Pettersen, C. G. Fletcher, and A. Geiss, 2024: Development of a Full-Scale Connected U-Net for Reflectivity Inpainting in Spaceborne Radar Blind Zones. Artif. Intell. Earth Syst., 3, e230063, https://doi.org/10.1175/AIES-D-23-0063.1.

Lenhardt, J., Quaas, J., Sejdinovic, D., and Klocke, D.: CloudViT: classifying cloud types in global satellite data and in kilometre-resolution simulations using vision transformers, EGUsphere [preprint], https://doi.org/10.5194/egusphere-2024-2724, 2024.

Thisanke, H., Deshan, C., Chamith, K., Seneviratne, S., Vidanaarachchi, R., & Herath, D. (2023). Semantic segmentation using Vision Transformers: A survey. Engineering Applications of Artificial Intelligence, 126, 106669. https://doi.org/10.1016/j.engappai.2023.106669

---

## Author Response (AR2)

**Reply to Reviewer #1's comments**

I thank the authors for addressing some of my comments. I have a few minor points to follow-up on.

Answer: We sincerely thank the reviewer for the suggestions and continued engagement. In response, we revised the manuscript according to the reviewer's comments.

• First, it would be helpful for readers to understand the limitations of the dataset in the Abstract, in particular, in regards to the lack of use of lidar data for liquid classifications. I recommend that the author include this important information in the Abstract.

Answer: We appreciate the reviewer's suggestion. In response, we have added a sentence to lines 24–25 of the Abstract: 'Lidar measurements exhibit lower feature importance due to rapid signal attenuation caused by the frequent presence of persistent low-level clouds at the NSA site.'

• Second, the responses indicate that the references in the introduction were added but they were not.

Answer: We apologize for the oversight in the previous revision. The missing references cited in the Introduction section have now been properly included and verified.

• Third, I would appreciate if the authors could examine a few single-layer cloud cases as I previously suggested, even if in the SI section.

Answer: We have addressed the reviewer's concern in an additional figure in the supplement, S3, and added a sentence to lines 435-436 in the main body of the text: "Lidar measurement's lower feature importance in deference to radar measurements was also observed on days with single-layer, low-level liquid clouds (Figure S3)".

Note to the editor, we have rearranged the numbering on our supplementary figures.